# Exploring roles of the chitinase ChiC in modulating *Pseudomonas aeruginosa* virulence phenotypes

Per Kristian Thorén Edvardsen,[1] Fatemeh Askarian,[2] Raymond Zurich,[2] Victor Nizet,[2,3] Gustav Vaaje-Kolstad[1]

**ABSTRACT** Chitinases are ubiquitous enzymes involved in biomass degradation and chitin turnover in nature. *Pseudomonas aeruginosa* (PA), an opportunistic human pathogen, expresses ChiC, a secreted glycoside hydrolase 18 family chitinase. Despite speculation about ChiC's role in PA disease pathogenesis, there is scant evidence supporting this hypothesis. Since PA cannot catabolize chitin, we investigated the potential function(s) of ChiC in PA pathophysiology. Our findings show that ChiC exhibits activity against both insoluble (α- and β-chitin) and soluble chitooligosaccharides. Enzyme kinetics toward $(GlcNAc)_4$ revealed a $k_{cat}$ of 6.50 $s^{-1}$ and a $K_M$ of 1.38 mM, the latter remarkably high for a canonical chitinase. In our label-free proteomics investigation, ChiC was among the most abundant proteins in the Pel biofilm, suggesting a potential contribution to PA biofilm formation. Using an intratracheal challenge model of PA pneumonia, the *chiC*::ISphoA/hah transposon insertion mutant paradoxically showed slightly increased virulence compared to the wild-type parent strain. Our results indicate that ChiC is a genuine chitinase that contributes to a PA pathoadaptive pathway.

**IMPORTANCE** In addition to performing chitin degradation, chitinases from the glycoside hydrolase 18 family have been found to play important roles during pathogenic bacterial infection. *Pseudomonas aeruginosa* is an opportunistic pathogen capable of causing pneumonia in immunocompromised individuals. Despite not being able to grow on chitin, the bacterium produces a chitinase (ChiC) with hitherto unknown function. This study describes an in-depth characterization of ChiC, focusing on its potential contribution to the bacterium's disease-causing ability. We demonstrate that ChiC can degrade both polymeric chitin and chitooligosaccharides, and proteomic analysis of *Pseudomonas aeruginosa* biofilm revealed an abundance of ChiC, hinting at a potential role in biofilm formation. Surprisingly, a mutant strain incapable of ChiC production showed higher virulence than the wild-type strain. While ChiC appears to be a genuine chitinase, further investigation is required to fully elucidate its contribution to *Pseudomonas aeruginosa* virulence, an important task given the evident health risk posed by this bacterium.

**KEYWORDS** *Pseudomonas aeruginosa*, chitinase, GH18, virulence, biofilm

Address correspondence to Per Kristian Thorén Edvardsen, per.kristian.edvardsen@nmbu.no, or Gustav Vaaje-Kolstad, gustav.vaaje-kolstad@nmbu.no.

The authors declare no conflict of interest.

See the funding table on p. 24.

*P*seudomonas aeruginosa (PA) is a Gram-negative bacterium widely distributed in nature (1, 2). It is capable of infecting a diverse range of organisms, including humans, various other mammals and vertebrates, plants, and insects (2, 3). Recognized by the World Health Organization as one of the top three critical bacteria urgently requiring new antibiotic treatments (4, 5), PA is a major pathogen linked to pneumonia in immunocompromised individuals, particularly cystic fibrosis patients. This association contributes to a decline in lung function and, in severe cases, death (2, 3). Given the evident health risk posed by PA in contemporary society, a more comprehen-

sive understanding of the mechanisms underlying the bacterium's pathogenicity and virulence is imperative.

Decades of research have focused on uncovering the virulence determinants of PA, with many of these mechanisms well studied. Numerous virulence factors, including enzymes secreted by the bacterium, have been identified. These include proteases such as elastases, which inflict damage on host tissues and deactivate components of the immune system; exotoxin A, which inhibits host protein synthesis; and lytic polysaccharide monooxygenase CbpD, recently identified to impair complement-mediated killing of PA (6–10). Additionally, PA secretes a glycoside hydrolase 18 (GH18) family chitinase called ChiC (11, 12). While chitinases are typically associated with chitin metabolism, this function seems unlikely for the PA chitinase, as the bacterium cannot degrade or utilize chitin as a nutrient source (11, 12). Despite this, ChiC has been associated with PA pathogenicity in various studies (13–15); however, direct evidence supporting an alternative function is scarce.

In other pathogens, alternative roles for chitinases have been elucidated. Examples include ChiA from *Listeria monocytogenes* that suppresses host innate immune cell function by lowering the expression of inducible nitric oxide synthase (16), ChiA2 from *Vibrio cholerae* that deglycosylates mucins and promotes intestinal colonization (17), ChiA from *Legionella pneumophila* that binds mucins and facilitates bacterial transit through the alveolar mucosa (18), and ChiA from *Salmonella Typhimurium* that remodels the intestinal glycome to promote small intestinal invasion (19). As each of these bacterial chitinases plays crucial roles in pathogenesis, gaining a deeper understanding of ChiC's molecular function(s) is essential to uncover its role in disease progression.

In this study, we present evidence showing the properties of ChiC for binding and hydrolyzing chitin particles as well as water-soluble chitooligosaccharides. Our findings indicate that PA exhibits only limited growth on chitin breakdown products and displays a low affinity for chitotetraose $(GlcNAc)_4$. Furthermore, we demonstrate the protein's ability to bind to a mucin-enriched extract, inducing chitinase synthesis *in vitro*. Furthermore, we showed that the protein could bind to a mucin-enriched extract, which also triggered chitinase production in vitro. Through a combination of label-free proteomics and an in vivo mouse infection model experiment, we were able to demonstrate that PaChiC is the most abundant protein in the Pel biofilm, and that the *chiC*::ISphoA/hah transposon insertion mutant showed marginally enhanced PA virulence. Our work establishes ChiC as a hydrolase proficient in cleaving β-1,4-linked *N*-acetylglucosamine moieties, emphasizing its contribution to PA pathoadaptive processes distinct from chitin degradation.

## RESULTS

### Sequence analysis and structure prediction

Analysis of the ChiC (PAO1, PA2300) amino acid sequence (UniProt ID: Q9I1H5) using InterPro (20) and the predicted AlphaFold2 structure (Fig. S1) revealed a tertiary structure comprising three domains: a glycoside hydrolase 18 domain (GH18: residues 15–335), a fibronectin type III domain (FnIII: residues 341–430), and a carbohydrate-binding module 5 or 12 (CBM5/12: residues 436–483) (Fig. 1A). Although no apparent signal peptide was identified for the protein using the SignalP-6.0 server (21), it is notable that the first 11 amino acids have been reported to undergo proteolytical processing in earlier studies of ChiC (12, 22). According to the AlphaFold2 (23) structure prediction, the GH18 domain adopts the typical TIM barrel fold of the GH18 family and contains the conserved DXXDXDXE motif, with Glu143 identified as the catalytic residue (Fig. S2 and S4G through I). The FnIII-like domain, commonly identified in bacterial carbohydrate-active enzymes, plays an unclear role. The CBM5/12 domain is known to bind chitin chains through surface-exposed aromatic amino acids (24–26).

Proteins structurally similar to ChiC were identified using the Dali server (28), revealing the endo-chitinase ChiC from *Serratia marcescens* (SmChiC; PDB ID: 4AXN) and the chitinase of *Moritella marina* (PDB ID: 4MB4) as the closest structural homologs (α

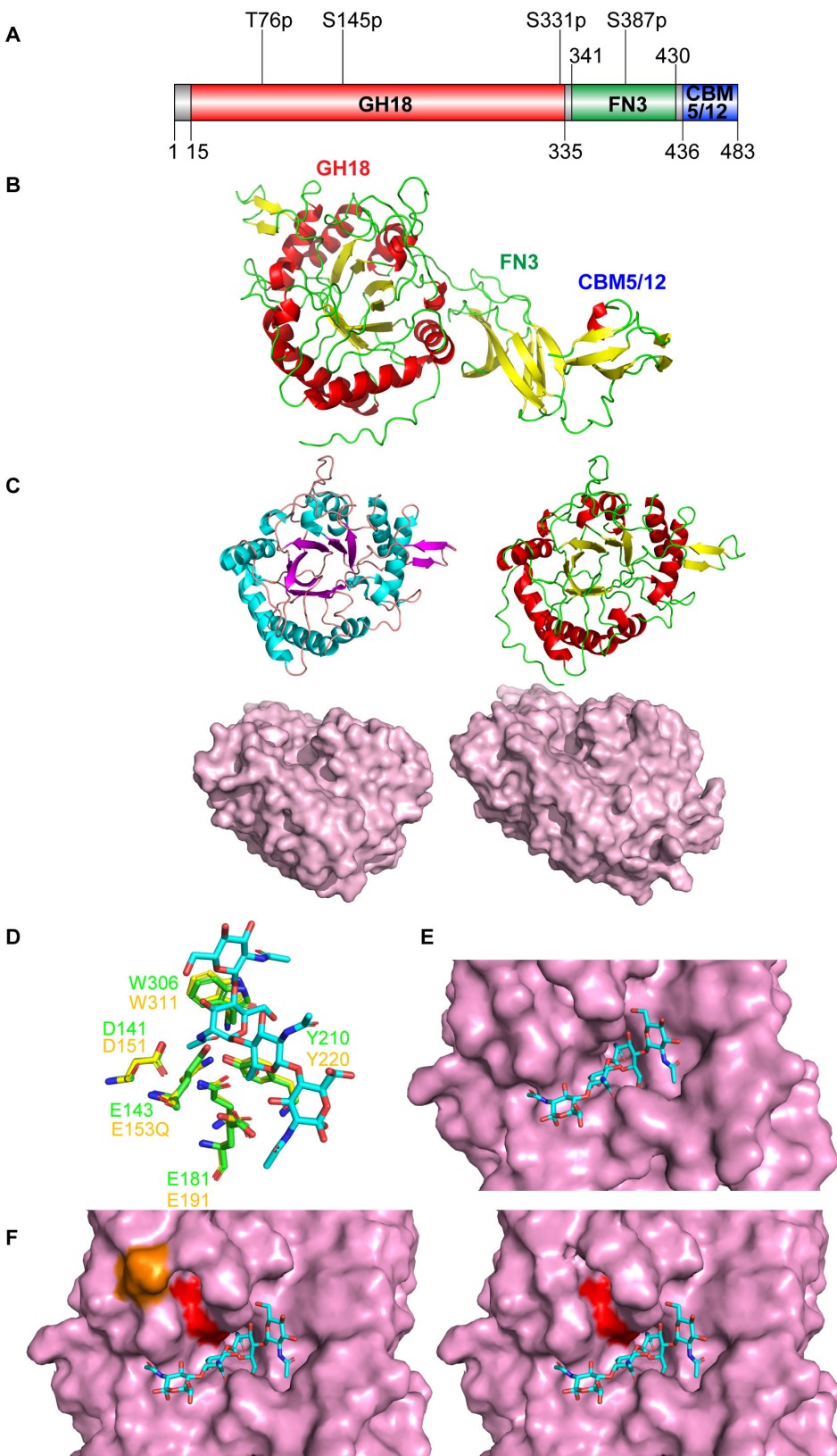

**FIG 1** ChiC sequence analysis and tertiary structure prediction. (A) Domain architecture of ChiC. The full-length protein comprises three domains: a glycoside hydrolase 18 domain (GH18), a fibronectin type III domain (FN3), and a carbohydrate binding module 5 or 12 (CBM5/12). Potential post-translational modifications (PTMs) are depicted above (27). The first

**FIG 1 (Continued)**

character is the amino acid followed by the amino acid number and the type of PTM. All postulated PTMs are phosphoryla-tions (p). (B) Model of ChiC predicted by AlphaFold2 provided by UNINETT Sigma2. (C) Comparison of the structures of the GH18 domains of *Sm*ChiC (PDB: 4AXN) (left) and ChiC (right). The GH18 domains are shown as both cartoons and surface models. The surface models accentuate the binding clefts of the GH18 domains. (D) Comparison of the active sites of ChiC with the chitinase of *Moritella marina* E153Q mutant in complex with (GlcNAc)$_4$ (PDB ID: 4MB4). (E) Showing (GlcNAc)$_4$ (PDB ID: 4MB4) superimposed into the active site of ChiC. (F) Comparison of the active site of the GH18 domain of ChiC with (GlcNAc)$_4$ (PDB ID: 4MB4), with (orange) and without phosphorylated serine at position 145. The catalytic glutamate at position 143 is highlighted in red. The addition of the phosphate group to serine was done using the plugin PyTMs version 1.2 in PyMol.

-carbon root-mean-square deviation of 0.5–0.6 and 0.8 Å and *Z*-scores of 54.3 and 51.0, respectively). The AlphaFold2 predicted structure of ChiC was analyzed and compared to the structure of *Sm*ChiC (64% protein identity) (Fig. S2; Fig. 1C). The analysis revealed substantial comparability in the catalytic GH18 domains, featuring a shallow/open substrate binding cleft indicative of endo-activity or enzymes that cleave intricate glycan structures (Fig. 1C). Additionally, analysis and comparison of the predicted structure of ChiC were conducted with that of *M. marina's* chitinase, for which an enzyme–substrate complex is known (PDB ID 4MB4; Fig. 1D). The key residues in both active sites are identical, and the (GlcNAc)$_4$ molecule in the *M. marina* enzyme is highly compatible with the active site of ChiC (Fig. 1E). Notably, when produced by PA, Ser145 (~10 Å from the Glu143 side chain), an amino acid close to the active site, has been reported to be phosphorylated (27), potentially influencing substrate specificity and enzyme activity (Fig. 1F).

## ChiC hydrolyses chitin particles and chitooligosaccharides

Considering its high structural resemblance to *Sm*ChiC, we investigated the activity of ChiC toward α-chitin and β-chitin. The protein bound and hydrolyzed both types of chitin, displaying a preference for β-chitin (Fig. 2A; Fig. S3). This confirms the chitino-lytic activity of the enzyme, with the primary product identified as (GlcNAc)$_2$ (Fig. 2B), consistent with the typical product pattern for GH18 chitinases. In the chitin degradation assay, β-chitin yielded 36-fold more product (GlcNAc)$_2$ compared to α-chitin after a 2 h incubation, as was anticipated due to the less recalcitrant nature of the β-chitin allomorph (Fig. 2B; controls and reaction product analysis are shown in Fig. S4A through F). In an activity assay comparing ChiC to *Sm*ChiC, the latter exhibited approximately twofold higher hydrolytic activity in the first 4 h of the reaction, followed by roughly equal activity for the remaining duration of the assay (Fig. 2C).

The kinetic properties of ChiC were determined using the soluble chitinous substrate (GlcNAc)$_4$. Analysis of the Michaelis–Menten kinetics revealed a $k_{cat}$ of 6.5 s$^{-1}$ (95% CI: 5.2–8.7 s$^{-1}$) and a $K_M$ of 1.4 mM (95% CI: 0.7–2.8 mM) (Fig. 2D; graphs displaying velocities shown in Fig. S5A through I). Notably, the $K_M$ value is significantly higher than what is reported for other endo-chitinases, e.g., the *Penaeus japonicus* endo-chitinase shows a $K_M$ of 249 µM for chitotetraose (29) and the *S. marcescens* endo-chitinase *Sm*ChiC shows a $K_M$ of 80 µM for the chitotetraose analog 4MU-(GlcNAc)$_3$ (30). This suggests that ChiC may have evolved to bind and/or cleave other substrates. This was further investigated experimentally by employing a glycan array binding analysis, using the inactive variant of ChiC (ChiC$_{E143Q}$; mutation of the catalytic glutamate), to prevent hydrolysis of potential substrates. The glycan array of (mostly) mammalian glycans, containing 585 glycan structures, it was revealed that ChiC$_{E143Q}$ only exhibited strong binding to (GlcNAc)$_5$ and (GlcNAc)$_6$, both oligosaccharides of chitin, with low affinity to other oligosaccharides (Table S1).

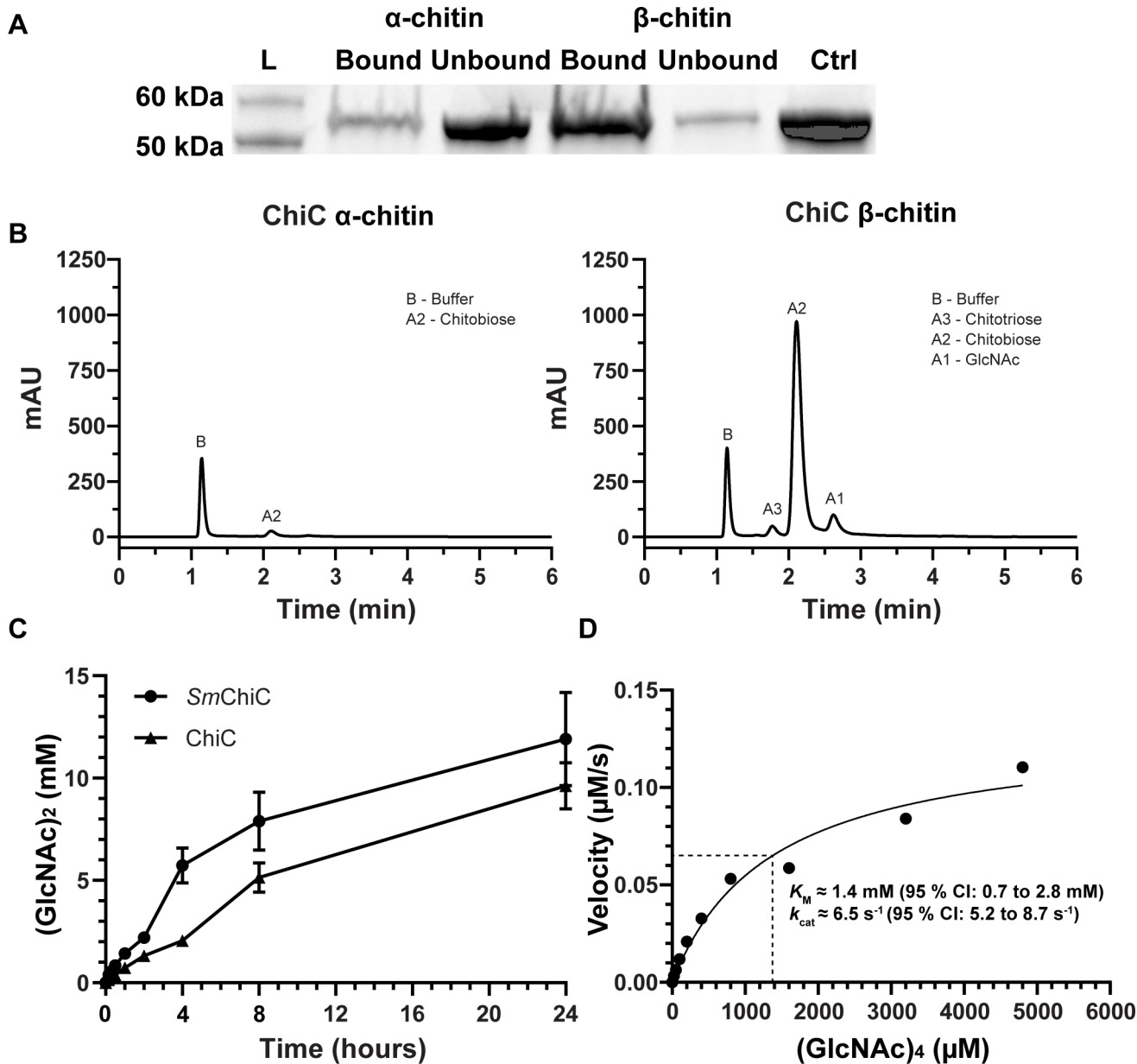

**FIG 2** ChiC binding and activity against chitin and chitooligosaccharides. (A) SDS-PAGE analysis showing the bound and unbound fractions of ChiC to α- and β-chitin, including a control with only ChiC. (B) Chromatograms showing product formation after incubation of ChiC with α-chitin (left) and β-chitin (right) for 2 h at 37°C, pH 7.5. (C) The chitinase activity of *Sm*ChiC and ChiC was measured over a 24-h period at 37°C, pH 7.5. The data are plotted as the mean ± standard deviation (SD), representing experiments performed in triplicate. (D) Michaelis–Menten kinetic analysis of ChiC incubated with varying concentrations of $(GlcNAc)_4$ at 37°C. The velocity was determined by quantifying the cleavage of $(GlcNAc)_4$ into $(GlcNAc)_2$ over time using 20 nM ChiC. $K_M$ ([S] at 1/2 Vmax) is indicated by the dashed lines. The curve is fitted using the Michaelis–Menten model with non-linear regression in GraphPad Prism version 10.0.2 and shown as the best fit with a 95% confidence interval (CI).

## *P. aeruginosa* is not able to utilize $(GlcNAc)_2$, GalNAc, or GalN as nutrient sources

The metabolic capacity of PA to utilize the degradation products of chitin was evaluated, considering that the chitin monomer, GlcNAc, found in mucins and glycosaminoglycans, can promote PA virulence (31). Growth curves indicate that PA can utilize GlcNAc as a carbon source (32), albeit not as efficiently as glucose (Fig. 3A and B). Interestingly, the bacterium was not able to utilize $(GlcNAc)_2$ as a carbon source (Fig. 3C), nor was it able to

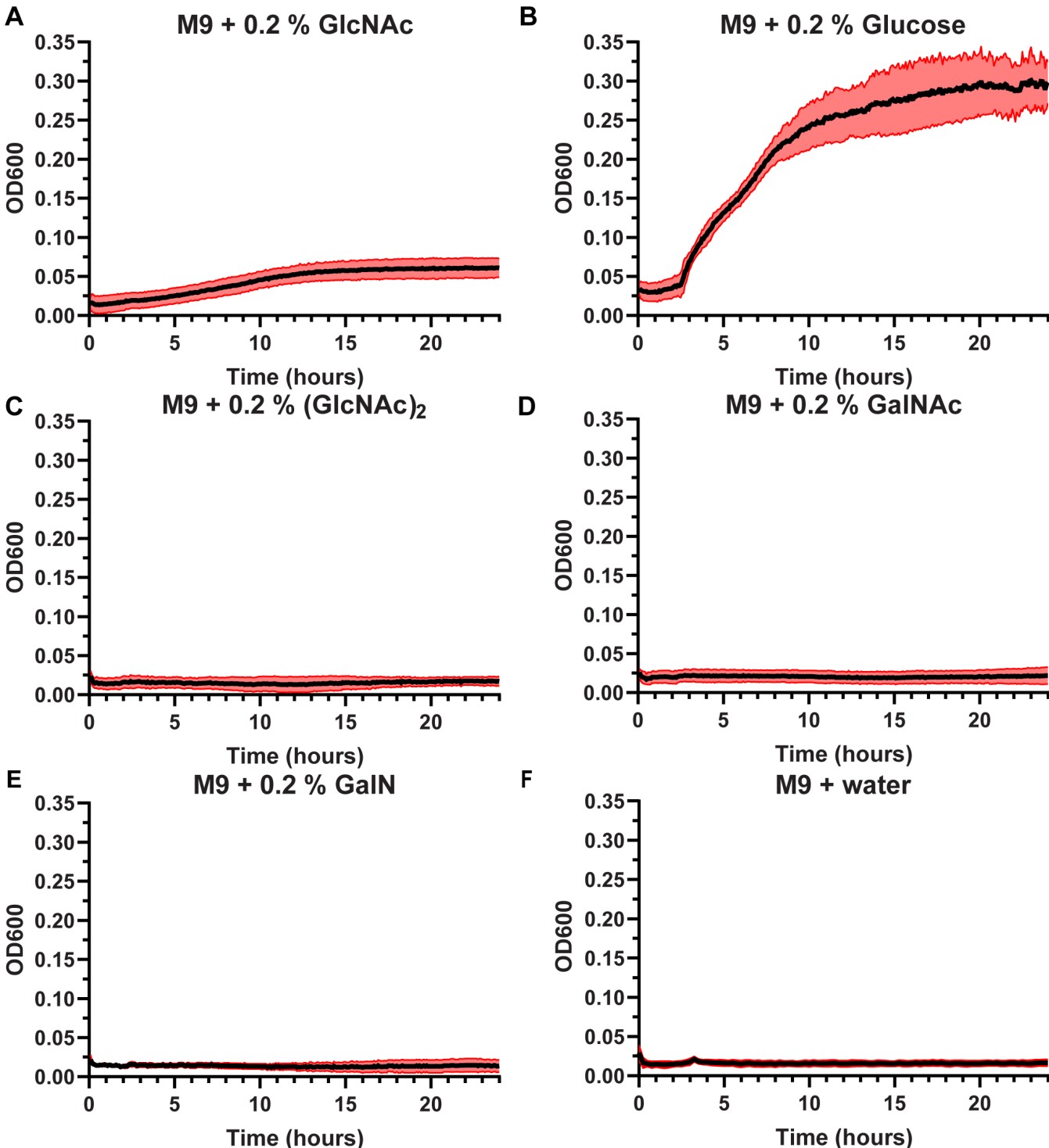

**FIG 3** Growth curves of PA wild type (WT) (PAO1) in M9 minimal medium with different carbon sources. Conditions evaluated were M9 medium supplemented with (A) 0.2% *N*-acetyl glucosamine (GlcNAc), (B) 0.2% glucose, (C) 0.2% *N*-acetyl chitobiose (GlcNAc)$_2$, (D) 0.2% *N*-acetyl galactosamine (GalNAc), (E) 0.2% galactosamine (GalN), and (F) water. The data are represented as the mean of four biological experiments, the black line, with the SD shown as the red area of the mean.

utilize GalNAc or GalN (Fig. 3D and E), two other amino sugars associated with PA pathogenicity. No growth was observed in the absence of carbon supply (Fig. 3F).

## Proteomic profiling of PA cultures spiked with $(GlcNAc)_2$

Given that PA cannot grow on $(GlcNAc)_2$ and considering that this disaccharide is the primary product generated by ChiC during incubation with β-chitin and chitin oligo-saccharides, we hypothesized that $(GlcNAc)_2$ could potentially function as a signaling molecule for the bacterium. With the knowledge that chitin and chitin oligosaccharides induce the type VI secretion system (T6SS) and competence in *Vibrio cholerae* (33), we reasoned that $(GlcNAc)_2$ might exert a regulatory effect on PA as well.

To explore this hypothesis, we analyzed the proteomic response of PA (UBCPP-PA14) cultivated in tissue-mimicking media (RPMI 1640) supplemented with 10% Luria broth (LB) to promote PA growth. $(GlcNAc)_2$ was introduced into the experimental conditions either 2 h (exponential growth) or 6 h (stationary phase) post-initial inoculation (Fig. 4A). The samples were collected 20 min after the administration of $(GlcNAc)_2$.

The data revealed that the administration of chitobiose induced a relatively substantial proteome alteration when introduced during the exponential phase, whereas a more modest response was observed when administered during the stationary phase of PA growth (Fig. 4B). Specifically, among the 2,949 proteins detected (Table S2), 195 and 12 proteins were significantly regulated during the exponential and stationary phases, respectively (Fig. 4B; Table S3). Notably, ChiC was not significantly regulated in either phase, with a $log_2$ fold change of 0.45 and 0.20, respectively. For Fig. 4C, we have only included proteins that have been previously associated with infection and virulence in the literature. By examining the string analysis network (Fig. 4D;Left cluster), one can observe the clustering of the upregulated proteins depicted in Fig. 4C, with the exception of ApeB, which was not found to be linked to any other proteins.

Given that quorum sensing tightly regulates many of the identified proteins in our analysis, we propose that $(GlcNAc)_2$ sensing serves as an alternative form of regulation for the genes encoding these proteins. Notably, no competence-related proteins were found to be modulated in our data set. Ultimately, the data point to a metabolic shift characterized by the downregulation of multiple ribosomal proteins and tRNA synthe-tases (Table S3), alongside an increased expression of membrane-associated proteins, DNA-binding proteins, and proteins associated with electron transport (e.g., FdnH, FdnG, SdhA, SdhB, Cc4, PA14_57570, and PA14_25840) (Fig. 4E; Table S3). During the stationary phase, only a few proteins were significantly regulated, with MttC, GlgB, and PopD found to be among the most significantly upregulated proteins (Fig. 4F).

## ChiC is the predominant protein in the Pel biofilm of PA

The Pel exopolysaccharide produced by PA consists of a dimeric repeat of α-1,4-linked GalN and GalNAc (34). The structure shares some resemblance to chitin through its linear structure and presence of 1,4-linked amino sugars. Earlier studies have indicated the presence of GlcNAc in Pel (35), although this monosaccharide was not identified in the recent study on the polysaccharide (34). Nonetheless, we found it relevant to investigate whether ChiC is related to Pel production and/or modification, especially since the enzyme may have evolved new activities and/or binding partners for the CBM5/12. In this regard, we examined the proteome of the Pel biofilm (proteins extracted from the pellicle) of PA14, a strain exclusively capable of Pel production and not of the other PA exopolysaccharides Psl and alginate (36, 37). The proteomic analysis identified a total of 2,418 proteins in the Pel biofilms (Table S4). Three proteins from the Pel machinery were detected, namely, PelA, PelB, and PelC, indicating the active production of Pel by PA (Table S4). Interestingly, ChiC was found to be the protein with the highest intensity on average for the three biological replicates (Table 1; Fig. 5A), establishing a functional connection to the Pel biofilm.

The proteomic identification of ChiC in the Pel biofilm cannot pinpoint whether the enzyme is intracellular or adhered to the biofilm matrix. To further investigate this aspect, we conducted an experiment in which we incubated a thoroughly washed Pel pellicle sample with the model substrate 4MU-$(GlcNAc)_2$. The efficient hydrolysis of the chitotriose analog (Fig. 5B) suggested that the entire enzyme or parts of it produced

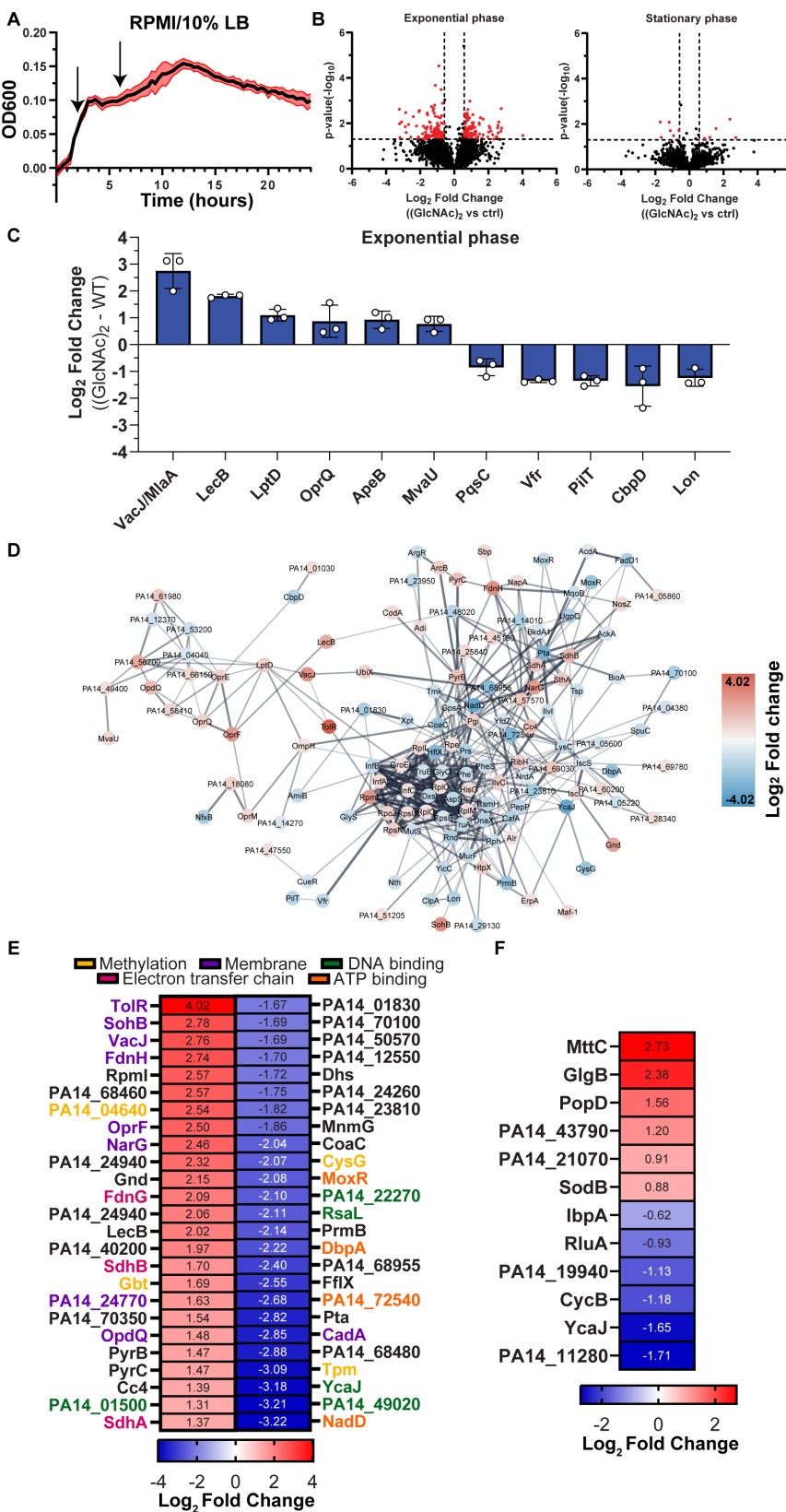

**FIG 4** Proteomic analysis of PA (WT) (UCBPP-PA14) spiked with (GlcNAc)$_2$. (A) Growth curve of PA WT (UCBPP-PA14) in RPMI 1640 supplemented with 10% LB. The administration of (GlcNAc)$_2$ is indicated by arrows. (B) Volcano plots displaying the log$_2$ fold change of each detected protein and their

**FIG 4** (Continued)

corresponding $P$-values ($-\log_{10}$) within the proteomics data set. The plots compare the $(GlcNAc)_2$ against the control (ctrl) treatment (addition of water) when supplemented at exponential and stationary growth phases. Significance was determined by a paired two-tailed $t$-test, and the cutoff was defined as $P = 0.05$ ($-\log_{10} = 1.3$) and ($\pm$)1.5-fold change ($\log_2 = 0.58$). (C) Histogram displaying the $\log_2$ fold change of significant up- and downregulated proteins associated with infection and virulence. The data are plotted as the mean $\pm$ SD. (D) STRING network analysis illustrating the interconnections of the significantly upregulated (red) and downregulated (blue) proteins when $(GlcNAc)_2$ was administered at exponential phase of PA growth. The average fold change ($\log_2$) is depicted within the nodes. Proteins without any connections to the network are omitted from the visualization (singletons). The default confidence cutoff of 0.4 was used for the network analysis. (E, F) A heatmap showing the top 25 up- and downregulated proteins in the samples comparing the $(GlcNAc)_2$ against control (ctrl) when supplemented at the exponential and stationary growth phases (marked with black square). The average $\log_2$ fold change is depicted. The proteins denoted by the various colors are associated with distinct GO terms.

by the bacterium are likely bound to the pellicle. Additionally, we tested the activity of the extracted Pel biofilms from the *chiC*::MAR2xT7 transposon insertion mutants D7 and D11, previously described (53), on the 4MU-$(GlcNAc)_2$ substrate. We anticipated that the extracted biofilms would exhibit no chitinase activity toward this substrate, as both mutants harbor transposons within the gene encoding ChiC. Our results show that the D7 transposon mutant displayed no discernible activity against this substrate, whereas the D11 transposon mutant exhibited some degree of activity. The disparity in activity levels between the D7 and D11 transposon mutants may be attributed to the insertion of the transposon in distinct regions of *chiC*, allowing expression of partly active ChiC.

Having established a functional connection between ChiC and Pel, in addition to showing that the protein still possesses catalytic activity in the biofilm itself, we investigated whether the protein could be of importance for the formation of the Pel biofilm. To test this hypothesis, we decided to compare and determine the biofilm phenotypes of the wild type (WT) and *chiC*::MAR2xT7 D7 transposon mutant incubated statically at 20°C and 37°C for 1 week. Visual inspection of the biofilms revealed comparable phenotypes in terms of appearance and formation (Fig. S6 and S7). Thus, ChiC does not appear to be involved directly in the development of the Pel biofilm, although further studies would be required to confirm this hypothesis.

**TABLE 1** Overview of selected proteins detected in the Pel biofilm associated with either infection or biofilm development

| Locus tag | Protein name | Protein function | Mean $\log_2$ fold change (relative to NuoC) | Ref |
|---|---|---|---|---|
| PA14_34870 | ChiC | Not known, possibly involved in cleavage and/or binding to carbohydrates. | 5.50 | –[a] |
| PA14_26020 | Lap/PaAP/PepB | Secreted protease regulated by quorum sensing (LasRI). One of the most abundant proteins in the biofilm matrix and important for biofilm development. | 4.05 | (38, 39) |
| PA14_31290 | LecA (lectin A; PA-IL) | Adhesion, binding of host glycans and biofilm development. | 3.91 | (40, 41) |
| PA14_16250 | LasB (elastase) | Immune evasion, cleavage of immune effectors, and inhibition of immune cells. | 3.70 | (6, 7, 42) |
| PA14_65000 | Azu | Secreted blue copper bound protein involved in the copper homeostasis of *P. aeruginosa*. The protein has also been shown to induce apoptosis in macrophages and cancer cell lines. | 3.57 | (43–48) |
| PA14_61200 | CdrA | Extracellular adhesin promoting stability and aggregation of the biofilms Psl and Pel. | 3.54 | (49, 50) |
| PA14_53250 | CbpD | Functions in immune evasion. | 3.36 | (9, 10) |
| PA14_50290 | FliC | Flagellin, the minor subunit that polymerizes to form the flagella. | 3.29 | (51, 52) |

[a]Note that "–" represents a lack of available references pinpointing the specific function of the protein.

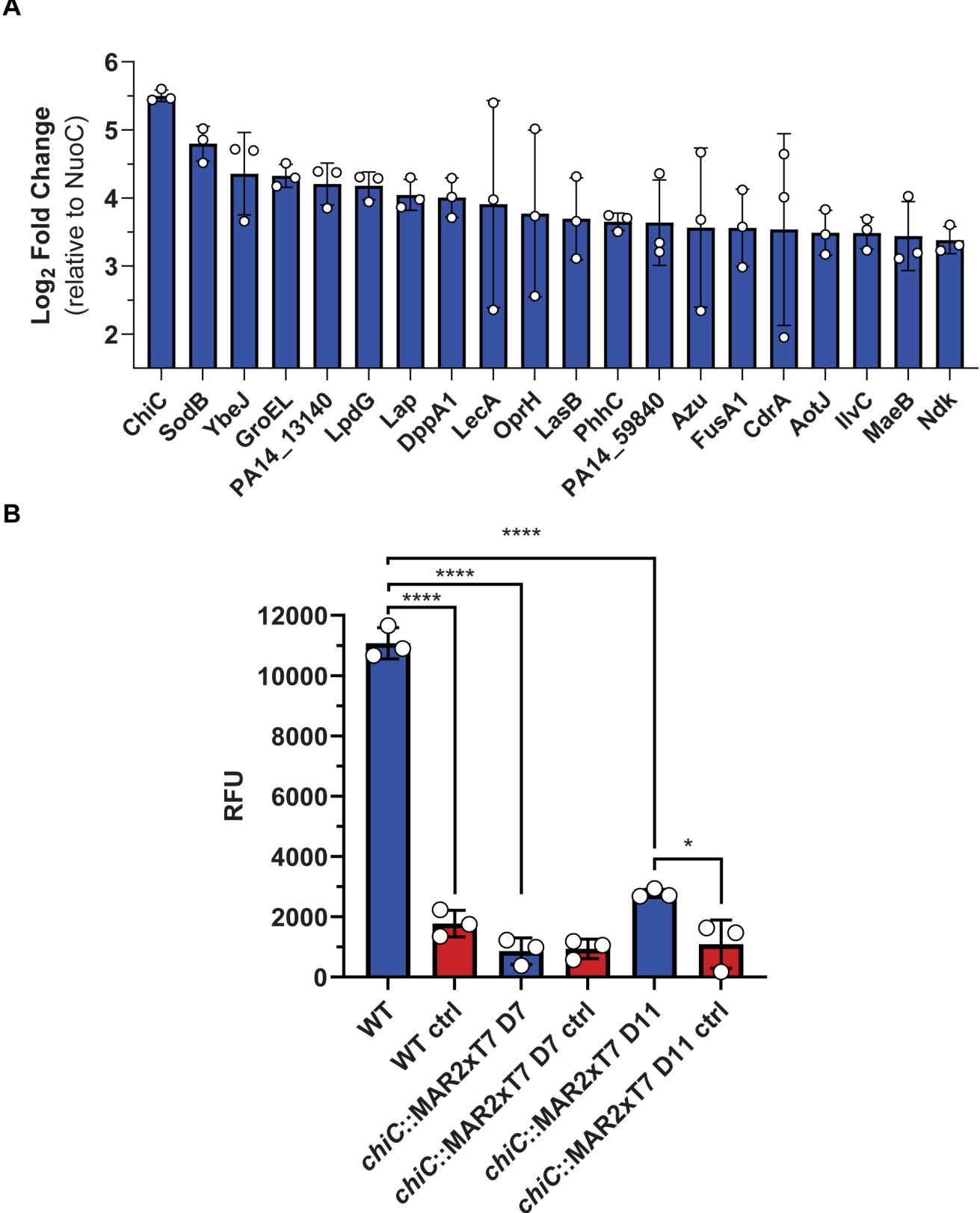

**FIG 5** The proteome and chinolytic activity of the Pel biofilm. (A) A histogram displaying the top 20 abundant proteins in the Pel biofilm presented as the mean $\log_2$ fold change ± SD relative to the housekeeping protein NADH-quinone oxidoreductase subunit C/D (NuoC). (B) Activity of the Pel biofilm against the substrate analog 4MU-$(GlcNAc)_2$, incubated for 2 h at 37°C. Water was added in the control samples replacing 4MU-$(GlcNAc)_2$. The measured relative fluorescence unit (RFU) is displayed as the mean ± SD, representing three biological replicates and two technical replicates. Data are analyzed by one-way ANOVA (Tukey's multiple comparisons).

## ChiC binds mucin from porcine stomach

Previous transcriptomic studies (54, 55) have demonstrated the upregulation of *chiC* in media supplemented with mucus. Given this and considering that other bacterial chitinases have been shown to adhere to various host glycoconjugates and mucin, we hypothesized that ChiC might interact with mucins. We investigated whether both ChiC and ChiC$_{E143Q}$ (inactive variant) could bind to immobilized mucin extract from porcine stomach (type III) using an ELISA assay. Interestingly, the results showed that both ChiC and ChiC$_{E143Q}$ were capable of binding to the mucin extract, although the inactive variant exhibited a stronger binding affinity than the wild-type enzyme (Fig. 6A). This prompted further investigation into the possibility that ChiC had a potential substrate in the mucin extract.

Given the ability of ChiC to bind mucin, we sought to determine whether the mucin extract could trigger enzymatic activity and induce the secretion of ChiC. This was investigated by measuring ChiC activity in either the cell-free culture supernatant or PA pellet (PAO1 and PA14). Bacteria were grown in LB and sampled at 2, 6, and 24 h, corresponding to the early exponential phase, mid-exponential phase, and late stationary phase for both bacteria, in the presence or absence of porcine mucin extract. Intriguingly, cells from PAO1 exhibited chitinase activity at the early and mid-exponential growth phases (Fig. 6B), while PA14 cells showed chitinase activity at all examined timepoints (Fig. S8). No chitinase activity could be detected in the cell-free supernatants (Fig. 6B; Fig. S8). Next, the induction of ChiC by mucins was assessed by growing the PAO1 strain in LB supplemented with 1% mucin from porcine stomach (type III) and sampling at 2, 6, and 24 h (the PA14 strain was not used due to its constitutive expression of ChiC). Strikingly, activity was observed at all examined timepoints, both in cell-free supernatants and PA pellets (Fig. 6C), clearly demonstrating that ChiC is induced in the presence of the mucin extract.

The investigation into the hydrolytic activity of ChiC toward mucin was motivated by accumulating evidence related to this complex glycoprotein. Analytical size exclusion chromatography (SEC) revealed a change in the elution profile of mucin extracts from porcine stomach (type II: crude extract; type III: partially purified extract) when incubated with ChiC. A control reaction where the ChiC was replaced with the inactive mutant ChiC$_{E143Q}$ showed a comparable alteration in the mucin elution profile, as for the wild type enzyme (Fig. S9 and S10). Therefore, it is not feasible to conclusively state that ChiC is active toward the mucin extracts.

## Contribution of ChiC to virulence of PA in murine model of pneumonia

ChiC has been described as a virulence factor (56, 57), and considering the importance of chitinases in pathogenicity (16–19), we conducted an evaluation of the impact of ChiC in a murine intratracheal (IT) infection model. Female CD1 mice (10 weeks old) were IT-infected with $1.5 \times 10^7$ CFU/mouse for both the WT PA strain (PAO1) and the *chiC*::ISphoA/hah transposon insertion mutant (PW4487). The mice infected with the *chiC* transposon insertion mutant experienced 100% mortality within 54 h post-infection, while 20% of the WT-infected mice survived (Fig. 7A). The median survival of the mice was determined to be 20 and 44 h for the *chiC*::ISphoA/hah transposon insertion mutant and the WT parent strain, respectively.

To gain a deeper understanding of the virulent nature of the *chiC*::ISphoA/hah transposon insertion mutant, we compared the proteome response of the transposon mutant to the WT strain in RPMI 1640 supplemented with 10% LB (a growth medium considered to mimic host nutrient-poor conditions) during the exponential growth phase. In total, 91 proteins were found to be significantly regulated in the *chiC*::ISphoA/hah transposon insertion mutant versus the WT strain (Fig. 7B; Table S5 and S6). The top three most significantly upregulated proteins were PA4925 (probable mechanosensitive channel), LecA (lectin A; PA-IL), and HcpA/B/C (Table S6). Interestingly, disruption of *chiC* led to the upregulation of multiple proteins associated with pathogenicity, such as components of different type VI secretion systems, components of type

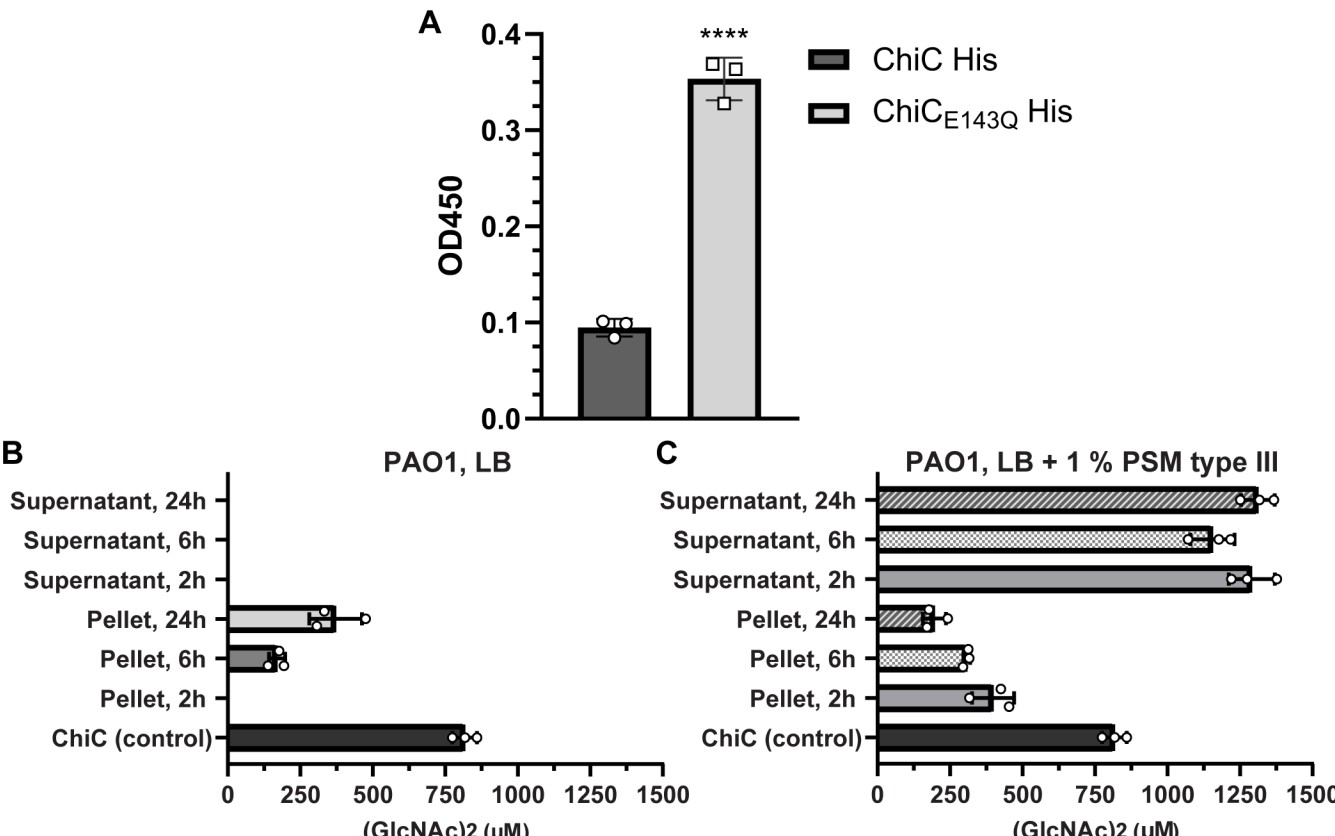

**FIG 6** Binding and induction of ChiC using mucin extract from porcine stomach. (A)ELISA analysis of binding between immobilized porcine stomach mucins type III and His-tagged ChiC (10 µM) and His-tagged ChiC$_{E143Q}$ (10 µM) detected with anti-His-tag antibody. $^{****}P < 0.0001$; ChiC$_{E143Q}$ versus ChiC by two-tailed Student's $t$-test. Buffer was used as a control and subtracted from the samples. Data plotted represented as the mean ± SD of three replicates. (B and C) Quantification of chitinase activity of PA (PAO1) at different growth phases in LB medium and LB medium supplemented with 1% PSM type III. Activity is represented by the concentration of generated (GlcNAc)$_2$ (µM) by the bacteria and the cell-free supernatants upon incubation with (GlcNAc)$_4$ for 2 h after the given growth phase was reached. Purified ChiC (1 µM) was used as a positive control. The data are plotted as the mean ± SD, representing three experiments performed in biological triplicates.

IV pili, and additional virulence factors (Fig. 7C) (6, 7, 40, 41, 58–65). Furthermore, the KEGG pathway enrichment analysis of the significantly upregulated proteins indicated that biofilm formation, quorum sensing, and bacterial secretion systems were among the pathways to be significantly enriched (Fig. 7D; Table S7). The increased expression of proteins involved in pathogenicity and enriched pathways, such as biofilm formation, quorum sensing, and bacterial secretion systems, suggests that the bacterium is actively attempting to recover from the inactivation and loss of ChiC.

## DISCUSSION

Enzymes within the GH18 family are primarily recognized for their roles in chitin depolymerization but have also been reported to hydrolyze fragments of peptidoglycan, Nod-factors, LacdiNAc, glycans from IgG and IgA, carbohydrates present on mammalian glycoproteins, and mucin extracts, illustrating the functional diversity of these enzymes (18, 66–74). In the context of chitin metabolism, PA is unable to grow on chitin particles (11), exhibits delayed growth on GlcNAc (Fig. 3A) (32), and cannot utilize (GlcNAc)$_2$ (Fig. 3C), the dominant degradation product of chitin hydrolysis by ChiC (Fig. 2B). Additionally, no fully sequenced PA strains contain genes encoding GH20 $N$-acetylglucosaminidases, which are required to hydrolyze (GlcNAc)$_2$ to two GlcNAc moieties. Based on these observations, it can be hypothesized that ChiC has evolved to target a substrate other than chitin or that the chitinolytic activity of the enzyme [releasing (GlcNAc)$_2$] is a

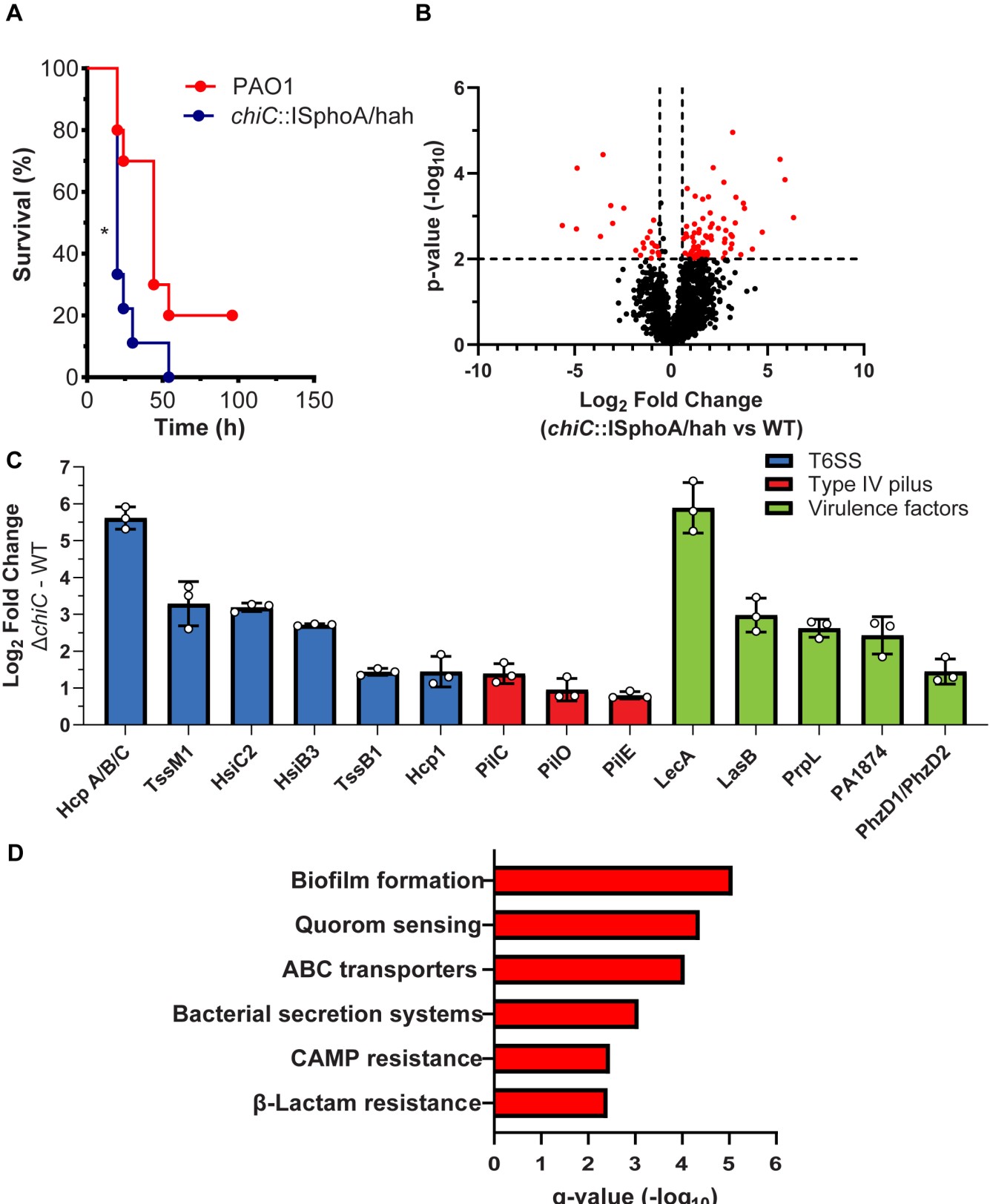

**FIG 7** Contribution of ChiC to virulence of PA. (A) Kaplan–Meier survival (%) curves of CD-1 mice inoculated with PAO1 and the *chiC*::ISphoA/hah transposon insertion mutant. The significance is indicated by an asterisk (*) ($P < 0.05$). $P = 0.022$ by the Mantel–Cox test. (B) Volcano plot showing the $\log_2$ fold change of all detected proteins and their corresponding $P$-values ($-\log_{10}$) in *chiC* transposon mutant against the WT parent strain. Significance was determined

**FIG 7** (Continued)

by a paired two-tailed $t$-test, and the cutoff was defined as $P = 0.01$ ($-\log_{10} = 2.0$) and ($\pm$)1.5-fold change ($\log_2 = 0.58$). (C) A histogram displaying selected significantly upregulated proteins associated with infection and virulence in the *chiC* transposon mutant against the WT strain. The average fold change ($\log_2$) is depicted $\pm$SD. (D) KEGG pathway enrichment analysis of the upregulated proteins comparing the *chiC* transposon mutant against the WT (red bars) with corresponding $q$-values ($-\log_{10}$). Proteins with $q$-values of $\leq 0.05$ were defined to be significant.

signaling event that triggers a specific response from the bacterium. The data obtained in the present study provide support for both hypotheses.

The substantial proteome alterations observed when a growing PA culture was spiked with $(GlcNAc)_2$ (Fig. 4B) lend credence to the latter theory. It is well established that $(GlcNAc)_2$ induces natural competence and expression of T6SS in *V. cholerae,* which is associated with chitin colonization (33). The present data do not indicate the induction of competence in PA, but rather the regulation of a variety of virulence factors. Although PA is an environmental bacterium, not much is known about its ability to interact with or degrade chitin. There is no obvious pathway involved in chitin degradation in PA, but the bacterium has been shown to utilize acetate from chitin degradation in co-culture with a chitinolytic bacterium (11). Because chitin does not occur naturally in humans, it raises the question of what the natural substrate for these enzymes is. One possibility can be mammalian glycans, as some of these contain $(GlcNAc)_2$, for example, in the core structures of *N*-glycans. Thus, it is not unlikely that the bacterium may encounter this dimeric sugar in a host environment, taking into account its binding and potential activity toward porcine gastric mucins (Fig. 6A; Fig. S9 and S10). While porcine gastric mucin serves as a beneficial model substrate, using human mucins would offer more relevant insights into the validity of the current findings.

The putative role of ChiC in the interaction of PA with mucus is supported by the enzyme being the dominant protein in the Pel biofilm matrix, as Pel has been shown to be important for cell aggregation, aminoglycoside resistance, and aggregation with sputum from cystic fibrosis patients (75). Porcine mucin indeed also triggered ChiC expression by PA (Fig. 6B and C; Fig. S8), a finding in line with transcriptomic studies by Cattoir et al. (55) and Devlin and Behnsen et al. (56) showing the upregulation of *chiC* in media containing mucus. Interestingly, a GH18 chitinase structurally similar to ChiC (Fig. 8) has been shown to possess an unprecedented $Zn^{2+}$-dependent peptidase activity enabling specific cleavage of the protein chain of the MUC5AC lung mucin (18). Therefore, a putative protease activity cannot be ruled out for ChiC. Indeed, such an activity might account for the comparable results observed between the ChiC wild type and the inactive E143Q variant in the mucin hydrolysis assay (Fig. S9 and S10). However, no activity was observed for ChiC towards casein (Fig. S11), but further experiments are required to pursue this hypothesis. Additionally, it is noteworthy that other GH18 family enzymes have been shown to have activity toward non-chitin substrates, such as ChiA from *S.* Typhimurium and *L. monocytogenes* (which cleaves LacdiNAc, a disaccharide motif common in mucins), ChiA from *L. pneumophila* (associated with mucin hydrolysis), and a recent study showing ChiA of *S.* Typhimurium (altering the *N*-glycome profile of gut epithelial cells) (18, 66, 67) (the structures of ChiC and these enzymes are shown in Fig. 8). Therefore, it is plausible that ChiC has assumed a similar function. In this study, the murine pneumonia model revealed higher virulence of the PA *chiC*::ISphoA/hah transposon insertion mutant compared to the wild-type parent strain, a finding consistent with the proteomic data. Such observations are not uncommon in experiments of this nature, as pathogens may adopt alternative survival strategies when specific virulence components are inactivated or absent (76, 77). Alternatively, the gene may have hindered the bacterium's virulence, suggesting a potential anti-virulence role, where its disappearance causes increased virulence of the organism (78–80). Nevertheless, more investigations are needed to validate the role of ChiC in the pathogenesis of PA.

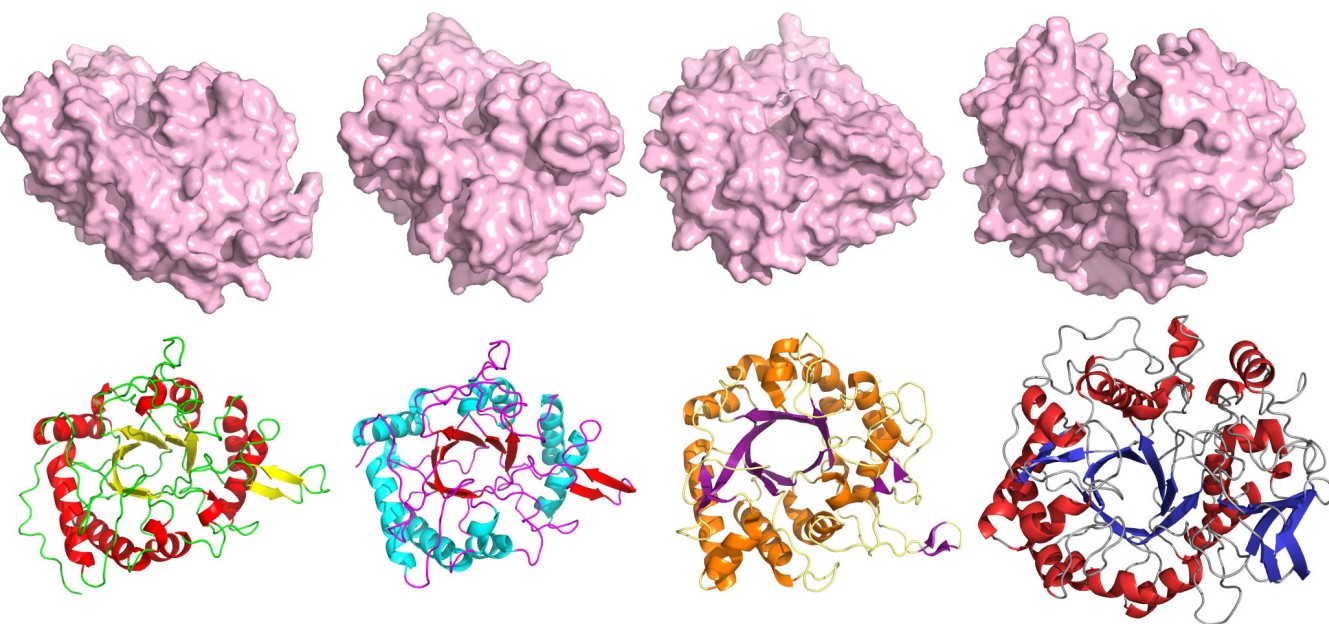

**FIG 8** Comparison of the GH18 domains of the chitinases from *PA*, *L. monocytogenes*, *L. pneumophila* (PDB: 6S2X), and *S. Typhimurium* (from left to right). The predicted structures from *L. monocytogenes* and *S. Typhimurium* were retrieved from the AlphaFold*2* database (UniProt entries Q8Y619 and Q8ZS09, respectively).

## MATERIALS AND METHODS

### Bacterial strains

*P. aeruginosa* strains used in the study include wild-type UCBPP-PA14 and PAO1, and the corresponding *chiC* transposon insertion mutants (Table 2). The commonly used BL21 Star (DE3) *E. coli* strain and *Vibrio natriegens* Vmax X2 were used for protein expression of ChiC and *Sm*ChiC, respectively (Table 2).

### Cloning

The chitinase (ChiC; PA2300) (UniProt ID; Q9I1H5) of *P. aeruginosa* PAO1 was synthesized and cloned into pNIC-CH (Addgene) using the GenScript gene cloning service. Two constructs were synthesized and cloned, one with and one without a hexahistidine tag attached to the C terminus of the protein. Both constructs were transformed and propagated in One Shot BL21 Star (DE3) *E. coli* cells (Invitrogen). A truncated version of ChiC was generated using the QuikChange II XL Site-Directed Mutagenesis Kit according to the manufacturer's instructions. The predicted catalytic residue glutamate was mutated to glutamine (E143Q).

### Expression and purification of ChiC from *E. coli*

The expression of native, His-tagged, and truncated versions of ChiC was performed by the cultivation of *E. coli* BL21 Star (DE3) containing the relevant plasmid in Terrific Broth medium supplemented with 50 µg/mL kanamycin. Isopropyl β-D-1-thiogalactopyranoside (IPTG) was added to a final concentration of 1 mM when the culture reached optical density (OD) of 0.5–0.7 measured at a wavelength of 600 nm. The culture was further incubated at 37°C for an additional 3 h before the pellets were harvested. Bacterial cells were harvested by centrifugation at 7,500 rpm for 15 min and resuspended in lysis/ binding buffer (5 mM imidazole, 150 mM NaCl, and 15 mM Tris-HCl pH 7.5), supplemented with Complete Mini EDTA-free protease inhibitors with a final concentration of 1× and a cocktail of phosphatase inhibitors: 1 mM beta-glycerophosphate (Sigma), 1 mM sodium orthovanadate (Sigma), and 10 mM sodium pyrophosphate (Sigma) followed by sonication using a Vibra-Cell Ultrasonic Processor (Sonics). The cells were sonicated for

**TABLE 2** Description and summary of the bacterial strains used in this study.

| Strain | Description | Ref |
|---|---|---|
| *Pseudomonas aeruginosa* UCBPP-PA14 | *Pseudomonas aeruginosa* UCBPP-PA14 | (81) |
| *Pseudomonas aeruginosa* UCBPP-PA14 *chiC*::MAR2xT7 D7 (PA14NR: 26353) | *Pseudomonas aeruginosa* UCBPP-PA14 transposon mutant, *chiC*::MAR2xT7. Features a transposon in *chiC*, resulting in the inactivation of the chitinase protein | (53) |
| *Pseudomonas aeruginosa* UCBPP-PA14 *chiC*::MAR2xT7 D11 (PA14NR: 42166) | *Pseudomonas aeruginosa* UCBPP-PA14 transposon mutant, *chiC*::MAR2xT7. Features a transposon in *chiC*, resulting in the inactivation of the chitinase protein | (53) |
| *Pseudomonas aeruginosa* PAO1 | *Pseudomonas aeruginosa* PAO1 | (82) |
| *Pseudomonas aeruginosa* PAO1 strain PW4887 (*chiC*::ISphoA/ hah) | *Pseudomonas aeruginosa* PAO1 transposon mutant, *chiC*::ISphoA/hah. Features a transposon in *chiC*, resulting in the inactivation of the chitinase protein | (82) |
| *Escherichia coli* BL21 Star (DE3) pNIC-CH ChiC (PA2300) | Strain for expression and purification of ChiC | This study |
| *E. coli* BL21 Star (DE3) pNIC-CH ChiC | Strain for expression and purification of ChiC with His-tag | This study |
| *E. coli* BL21 Star (DE3) pNIC-CH ChiC$_{E143Q}$ | Strain for expression and purification of ChiC with His-tag and mutated active site | This study |
| *Vibrio natriegens* Vmax X2 pET28b SmChiC | Strain for expression and purification of SmChiC | This study |

10 min using a cycle of 5 sec off and 5 sec on (30% amplitude). Cell debris was removed by centrifugation at 20,000 rpm for 15 min, and the cytoplasmic protein extract was filtered using a 0.2 µm filter.

Cytoplasmic extracts were loaded onto a HisTrap High-Performance column (Cytiva/GE Healthcare) connected to an ÄKTA pure protein purification system (Cytiva/GE Healthcare), and purification was performed based on the manufacturer's instructions.

Unwanted proteins were removed by gel filtration using an ÄKTA pure (Cytiva/GE Healthcare) operating a ProteoSEC Dynamic 16/60 3-70 HR size exclusion column (Protein Ark). The flow rate was set to 1 mL/min, and the buffer used was 15 mM Tris-HCl pH 7.5 with 150 mM NaCl. Fractions were pooled, concentrated, and buffer-exchanged into the same buffer used for the SEC column using a Vivaspin 20 (10 kDa molecular weight cutoff) centrifugal concentrator (Sartorius Stedim Biotech GmbH). Protein purity was estimated by SDS-PAGE to be >90%. Protein concentrations were determined using the theoretical extinction coefficient (96,260 M$^{-1}$ cm$^{-1}$) calculated by the ProtParam tool (http://web.expasy.org/protparam/) at 280 nm.

## Expression and purification of *Sm*ChiC from Vmax X2 cells

The expression of *Sm*ChiC was performed by the cultivation of Vmax X2 Chemicompetent Cells (BioCat GmbH, product number: CL1300-05-GVO-SGI) containing the plasmid pET28b construct with *Sm*ChiC (83). Overnight cultures were diluted 1/2,000 in a modified 2x YT medium (16 g/L tryptone, 10 g/L yeast extract, and 10 g/L NaCl) supplemented with 200 µg mL$^{-1}$ kanamycin in 1 L flasks at 30°C using a LEX-24 Bioreactor (Harbinger Biotechnology). IPTG was added to a final concentration of 1 mM when the culture reached OD600 of 0.5. The culture was further incubated at 30°C for an additional 20 h before the supernatant was harvested. Bacterial cells were separated from the supernatant by centrifugation at 7,500 rpm for 30 min. The supernatant was filtered twice using a 0.8 µm filter followed by a 0.45 µm filter. The supernatant was concentrated 10 times its original volume at 4°C using a Vivaflow 200 system (Sartorius Stedim Biotech GmbH) with a Masterflex Peristaltic Pump (Sartorius Stedim Biotech GmbH) and a Vivaflow Hydrosart Cassette, 30 kDa MWCO (Sartorius Stedim Biotech GmbH).

The supernatant was loaded onto two HiPrep 26/10 Desalting columns (Cytiva/GE Healthcare) connected in series using an ÄKTA pure protein purification system (Cytiva/GE Healthcare). The supernatant was desalted and buffer-exchanged into sterile filtered seawater as per the manufacturer's instructions. The protein was then purified using gel filtration, as previously described. Fractions were pooled, concentrated,

and buffer-exchanged using a Vivaspin 20 (10 kDa molecular weight cutoff) centrifugal concentrator (Sartorius Stedim Biotech GmbH). Protein purity was estimated by SDS-PAGE to be >90%. Protein concentrations were determined using the theoretical extinction coefficient (99,240 $M^{-1}$ $cm^{-1}$) calculated by the ProtParam tool (http://web.expasy.org/protparam/) at 280 nm.

## Bacterial growth curves

PAO1 was grown overnight in 4 mL LB at 37°C with shaking (200 rpm). The next day, the bacteria were pelleted by centrifugation for 5 min at $16,900 \times g$. The pellets were then resuspended in phosphate buffere saline (PBS) and centrifuged again. The supernatant was removed, and the pellets were resuspended in PBS. The bacteria were subsequently diluted to an OD600 of $\approx$ 0.05 (Eppendorf BioPhotometer) in M9 salts (Gibco) supplemented with 2 mM $MgSO_4$ and 0.1 mM $CaCl_2$, where the sole carbon source was glucose (Thermo Fisher Scientific), N-acetylglucosamine (GlcNAc) (Sigma, ≥95% purity), diacetyl-chitobiose $(GlcNAc)_2$ (Megazyme, ≥95% purity), N-acetylgalactosamine (Sigma, ≥98% purity), or galactosamine hydrochloride (Sigma, ≥99% purity) at a concentration of 0.2%. The bacteria were grown in a 96-well microtiter plate (total volumes of 150 µL) at 37°C. The growth was monitored by measuring the OD600 every 5 min using a Varioskan LUX multimode microplate reader (Thermo Fisher Scientific). Shaking was performed for 15 sec before each reading. The medium was included as a negative control. The data are from four biological replicates.

## ChiC binding to chitin

Reaction mixtures (50 µL) contained 20 mM Tris-HCl buffer pH 7.5 and 0.5 mg/mL of ChiC (E. coli) with α-chitin (final concentration 5 g/L) and β-chitin (final concentration 5 g/L). The reactions were incubated at 37°C for 1 h with shaking at 600 rpm in a Thermomixer C (Eppendorf, Hamburg, Germany). After incubation, the samples were centrifuged at $16,900 \times g$ for 5 min in an Eppendorf 5418R centrifuge. The supernatant was removed, and the pellet was washed once with PBS before the chitin was centrifuged again. The insoluble chitin was suspended in NuPAGE LDS sample buffer. Both the supernatant and the pellets were analyzed via SDS-PAGE gel.

## ChiC activity against β-chitin, α-chitin, and $(GlcNAc)_4$

Reaction mixtures (500 µL) contained 20 mM Tris-HCl buffer pH 7.5 and 1 µM ChiC or $ChiC_{E143Q}$ (E. coli) with α-chitin (final concentration 5 g/L), β-chitin (final concentration 5 g/L), or $(GlcNAc)_4$ (final concentration 1 g/L). The reactions were incubated at 37°C for 2 h with shaking at 600 rpm in a Thermomixer C (Eppendorf, Hamburg, Germany). After incubation, reactions were stopped by the addition of $H_2SO_4$ to a final concentration of 5 mM and centrifuged at $16,900 \times g$ for 5 min in an Eppendorf 5418R centrifuge. The obtained supernatant was filtered using a MultiScreenHTS HV Filter Plate 0.45 µm (Millipore). Product formation was assessed using a Dionex UltiMate 3000 HPLC system with UV light (194 nm). The samples were analyzed using the HPLC system with a 100 × 7.8 mm Rezex RFQ-Fast Acid H+ (8%) analytical column (Phenomenex, Torrance, CA, USA) operated at 85°C with 5 mM $H_2SO_4$ as the eluent and using an isocratic flow of 1.0 mL/min. The injection volume was set to 8 µL.

## Michaelis–Menten kinetics

Kinetic parameters of ChiC (E. coli) were determined using tetraacetyl-chitotetraose $(GlcNAc)_4$ (Megazyme, ≥95% purity) as the substrate. The reaction mixtures (250 µL) contained 20 mM Tris-HCl buffer pH 7.5 and 20 nM ChiC (E. coli), as well as varying concentrations of $(GlcNAc)_4$ ranging from 0 to 4.8 mM. The reactions were incubated at 37°C for 12 min with shaking at 600 rpm in a Thermomixer C (Eppendorf, Hamburg, Germany). Fifty-microliter samples were retrieved after 4, 8, and 12 min, and the reactions were stopped by adding $H_2SO_4$ to a final concentration of 5 mM. All reactions

were run in three parallels. The samples were filtered using a MultiScreenHTS HV Filter Plate 0.45 µm (Millipore). The quantification of product formation was performed as above, using a Dionex Ultimate 3000 HPLC system with a Rezex RFQ-Fast Acid H+ (8%) analytical column.

The kinetic parameters were obtained from triplicates of data fitting to the Michaelis–Menten equation via nonlinear regression using GraphPad Prism version 9.1.0 (GraphPad Software Inc, San Diego, CA).

## Product formation by β-chitin degradation

Reaction mixtures (800 µL) for the β-chitin degradation assay contained 20 mM Tris-HCl buffer pH 7.5, 1 µM ChiC (*E. coli*) or *Sm*ChiC (Vmax), and 10 g/L β-chitin. Reactions were incubated at 37°C for 24 h with shaking at 600 rpm in a Thermomixer C (Eppendorf, Hamburg, Germany). Fifty-microliter samples were retrieved after 10 min, 20 min, 30 min, 1 h, 2 h, 4 h, 8 h, and 24 h. The reactions were stopped by adding $H_2SO_4$ to a final concentration of 5 mM. All reactions were run in three parallels.

$(GlcNAc)_2$ formation by ChiC and *Sm*ChiC was quantified as described earlier, using a Dionex Ultimate 3000 HPLC system with a Rezex RFQ-Fast Acid H+ (8%) analytical column.

## Glycan microarray analysis

ChiC$_{E143Q}$ (*E. coli*) was screened for binding against a library containing 585 natural and synthetic mammalian glycans (version 5.4) at final concentrations of 5 and 50 µg/mL in replicates of six. The highest and lowest points from each set of six replicates were removed so the average is of four values rather than six to ensure the elimination of false hits. The results are presented as average RFUs. The screening was done by the Protein-Glycan Interaction Resource of the Consortium of Functional Glycomics (http://www.functionalglycomics.org).

## Proteome analysis of bacterial cultures supplemented with $(GlcNAc)_2$

Starter cultures of *P. aeruginosa* UCBPP-PA14 were grown in LB at 37°C with shaking overnight. The starter cultures were diluted 50 times in 10 mL RPMI supplemented with 10% LB and incubated for 2 or 6 h at 37°C with shaking at 200 rpm. After 2 or 6 h, the cultures were supplemented with diacetyl-chitobiose $(GlcNAc)_2$ (Megazyme, ≥95% purity) to a final concentration of approximately 1 mM or with the same volume of sterile water as a control. The cultures were further incubated for an additional 20 min at 37°C. Thereafter, phenylmethylsulfonyl fluoride (PMSF) (Sigma), PhosSTOP (Roche), and Complete Mini EDTA-free protease inhibitors (Roche) were added to the cultures to final concentrations of 1 mM, 1×, and 1×, respectively. The bacterial pellets and supernatants were separated by centrifugation (4,500 × *g*, 10 min, 4°C). The pellets were resuspended in lysis buffer containing 20 mM Tris-HCl (pH 7.5), 0.1 M NaCl, 1 mM EDTA, and 1× Complete Mini EDTA-free protease inhibitors. Cells were disrupted by sonication (10×, 5″ off–5″ on, 26% amp), and the cellular debris was cleared by centrifugation (16,900 × *g*, 5 min, 4°C). Protein extracts were then stored at −80°C.

Prior to protein digestion, the proteins from the extracts were precipitated using chloroform and methanol, essentially as described by Wessel and Flügge (84). For digestion, desalting, and cleaning up the extracted peptides, STraps were used (85). These were made accordingly: using an 18 g blunt-ended needle, four pieces of Empore C18 membrane (Sigma, catalog number 6683-U) were cut out. By a length of 1/32″ PEEKsil capillary or equivalent, the membrane pieces were pushed firmly into a 200 µL pipette tip. Twelve pieces of MK360 quartz filter (45 mm diameter) were also cut and pushed into the tip. The STraps were mounted onto LoBind tubes via holes in the lids, which were cut out beforehand. The protein extracts were boiled at 70°C for 10 min, and iodoacetamide (IAA) was added to the samples to a final concentration of 50 mM. The samples were incubated for 20 min in the dark. Following the IAA treatment, the

samples were acidified using phosphoric acid to a final concentration of 8.5% (vol/vol). To the STraps, 170 μL strapping solution (90% methanol, 50 mM pH 7.5) was added with the samples shortly thereafter being added to the top layer of the strapping solution. The tips were centrifuged for 10 min at 2,500 × g. The flowthrough was discarded, and 50 μL of strapping solution was added to the tips. The tips were centrifuged for 5 min at 2,500 × g. Again, the flowthrough was discarded, and 70 μL of ABC solution (0.1 M ammonium bicarbonate and 5% acetonitrile) was added to the tips, before being centrifuged for 5 min at 2,500 × g. The flowthrough was discarded, and 30 μL of ABC solution containing trypsin with a final concentration of 20 ng/μL was transferred to the tips. The solution was centrifuged for approximately 20 sec at 1,000 × g, leaving some of the liquid above the stacked filters. The tips were covered with parafilm and incubated at 47°C for 1 h. Next, the tips were centrifuged for 2 min at 4,000 × g. To the flowthrough, 50 μL 0.5% trifluoroacetic acid (TFA) was added, and the flowthrough/TFA solution was added back into the tips. The tips were centrifuged for 5 min at 2,500 × g before discarding the flowthrough. One hundred microliters of 0.1% TFA was added, and the tips were centrifuged for 5 min at 2,500 × g. The flowthrough was removed, and 50 μL of 80% acetonitrile containing 0.1% TFA was transferred to the tips to enable peptide elution. The solution was centrifuged for approximately 10 sec at 1,000 × g, and the tips were incubated at room temperature for 10 min. The tips were subsequently centrifuged for 10 min at 1,000 × g. The eluted peptides were evaporated using a SpeedVac system until dryness and then redissolved in 12 μL of a solution containing 0.05% TFA and 2% acetonitrile.

The peptide samples (200 ng) were analyzed by coupling a nano-UPLC (nanoElute, Bruker) to a trapped ion mobility spectrometry and a quadrupole time of flight mass spectrometer (timsTOF Pro, Bruker). Peptide separation was achieved using a PepSep 1.5 μm C18 reverse-phase 25 cm × 75 μm analytical column with a ZDV sprayer emitter and CaptiveSpray Insert (Bruker, Germany). The temperature of the column was kept at 50°C using the integrated oven. Equilibration of the column was performed prior to sample loading (equilibration pressure 800 bar). The flow rate was set to 300 nL/min, and the samples were separated using a solvent gradient from 5% to 25% solvent B over 70 min. After a 9 min ramp to 37% B, the gradient was ramped to 95% solvent B in 10 min and maintained at the same level for a final 10 min. In total, a run time of 99 min was used for the separation of the peptides. Solvent A consisted of 0.1% (vol/vol) formic acid in water and 99.9% water, while solvent B consisted of 0.1% (vol/vol) formic acid in water and 99.9% (vol/vol) acetonitrile.

The timsTOF Pro was run in the positive ion data-dependent acquisition PASEF mode with the control software Compass Hystar version 5.1.8.1 and timsControl version 1.1.19 68. The acquisition mass range was set to 100–1,700 m/z. The TIMS settings were 1/K0 start 0.85 V·s/cm$^2$ and 1/K0 end 1.4 V·s/cm$^2$, ramp time 100 ms, ramp rate 9.42 Hz, and duty cycle 100%. The capillary voltage was set at 1,400 V, dry gas at 3.0 L/min, and dry temp at 180°C. The MS/MS settings were the following: number of PASEF ramps 4, total cycle time 0.53 sec, charge range 0–5, scheduling target intensity 20,000, intensity threshold 2,500, active exclusion release after 0.4 min, and collision-induced energy (CID) ranging from 27 to 45 eV.

The raw files were processed using MaxQuant (version 2.2.0.0) for label-free quantification (LFQ) and searched *P. aeruginosa* strain UCBPP-PA14 (UniProt proteome: UP000000653), and the "match between runs" feature was applied using default parameters. Trypsin/P was set as the digestive enzyme, and a maximum of two missed cleavages were permitted. The peptides were filtered with a 1% level false discovery rate (FDR) using a revert decoy database. Carbamidomethylation of cysteines was set as a fixed modification, while protein *N*-terminal acetylation, oxidation of methionines, and deamidation of glutamines were included as variable modifications. For data analysis, Perseus version 2.0.7.0 was used, and the quantitative values were log$_2$-transformed. Valid values were filtered with a minimum of eight values in total, and missing values were imputed from a normally distributed curve of log$_2$-transformed values. The

significantly up- or downregulated proteins were determined by performing pairwise Student's $t$-test ($P = 0.05$). Differentially expressed proteins were defined by having $\log_2$ values of $\leq -0.58$ ($-1.5$) and $\log_2$ fold change of $\geq 0.58$ (1.5).

For the STRING analysis, the STRING app for Cytoscape (version 3.9.1) was used for visualizing the connections and the network between the significantly up- and downregulated proteins (86, 87). The significantly up- and downregulated proteins from strain UCBPP-PA14 were converted to the identifiers of the ortholog's proteins found in *P. aeruginosa* PAO1 before performing the search. The default confidence cutoff of 0.4 and *P. aeruginosa* PAO1 was used for the query search.

## Examination of the proteome of the Pel biofilm

Three biological replicates of *P. aeruginosa* UCBPP-PA14 were streaked out on LB agar plates and incubated at 37°C overnight. The next day, 75 mL of T-broth (10 g/L tryptone and 5 g/ NaCl) was inoculated with bacteria to an OD600 of 0.005 in 250 mL Erlenmeyer glass flasks and incubated statically for 1 week at 20°C. After 1 week of incubation, the pellicles formed were removed by swirling the sticky pellicle around a large 5 mL pipette tip and then pipetted carefully into a 15 mL falcon tube. Excess liquid from the culture was removed, if possible, by centrifugation at 4,700 rpm for 5 min at 4°C. 0.5 mL of lysis buffer (20 mM Tris-HCl pH 7.5, 100 mM NaCl, 1 mM EDTA, 1% SDS, and 1× Complete Mini EDTA-free protease inhibitors) was added to the pellicles, and the samples were vortexed for 1 min at max speed. The pellicles were subsequently disrupted by sonication (20×, 5″ off–5″ on, 26% amp), and the cellular debris was cleared by centrifugation (16,900 × $g$, 10 min, 4°C). Protein extracts were stored at −80°C.

Prior to protein digestion, the proteins from the extracts were precipitated using chloroform and methanol, essentially as described by Wessel and Flügge (84). The protein samples were in-gel-digested, and peptides were desalted and cleaned as previously described (88), with one small change: Imperial Protein Stain (Thermo Fisher Scientific, catalog number 24615) was used for staining the gel. Staining was performed for 1 h, and the gel was destained using water overnight at 4°C.

The same method and settings for the peptide analysis using the UPLC-timsTOF Pro system were used as described previously under Proteome analysis of bacterial cultures supplemented with $(GlcNAc)_2$ with a 100 min gradient, injecting 200 ng of peptides from the samples.

The raw files were processed as described earlier, with minor changes. The peptides were searched against the *P. aeruginosa* strain UCBPP-PA14 (UniProt proteome: UP000000653), and the "match between runs" feature was applied using default parameters. For data analysis, the quantitative values were $\log_2$-transformed, and valid values were filtered with a minimum of three values in total. The mean LFQ intensities were used for comparison of protein abundance.

## Activity of the Pel biofilm against 4MU-$(GlcNAc)_2$

Three biological replicates of *P. aeruginosa* UCBPP-PA14 and *chiC*::MAR2xT7 transposon mutants (D7 and D11) were streaked out on LB agar plates and incubated at 37°C overnight. The next day, 10 mL of T-broth (10 g/L tryptone and 5 g/ NaCl) was inoculated with bacteria to an OD600 of 0.005 in six-well plates and incubated statically for 1 week at 20°C. For all bacteria, three biological replicates and two technical replicates were grown in six-well plates. After 1 week of incubation, the pellicles were removed by swirling the sticky pellicle around a 1 mL pipette tip and then carefully transferred into a 1.5 mL Eppendorf tube. The Pel biofilm was washed once with PBS and then centrifuged at 10,000 rpm for 5 min at room temperature. The pelleted biofilms were subsequently incubated with 100 µL reactions of 20 mM Tris-HCl pH 7.5 and 10 µM 4MU-$(GlcNAc)_2$. Water was added to the control samples in place of substrate. The reactions were incubated at 37°C for 2 h and stopped by adding 50 µL of the samples to 1.95 mL of 0.2 M $Na_2CO_3$. The amount of 4MU released was measured using a Hoefer DQ300 Fluorometer (Hoefer Inc., Hill Road Holliston, USA) with UV Led, with excitation

in the range of 365–395 nm and emission in the range of 440–470 nm. One micromolar 4MU was used for calibration, and the calibration standard value was set to 500.

## Phenotypic analysis of the Pel biofilm

Four biological replicates of *P. aeruginosa* UCBPP-PA14 and *chiC*::MAR2xT7 transposon mutant (D7) were streaked out on LB agar plates and incubated at 37°C overnight. The next day, 1 mL of T-broth (10 g/L tryptone and 5 g/ NaCl) was inoculated with bacteria to an OD600 of 0.005 in 24-well plates and incubated statically for 1 week at 20°C and 37°C. For all bacteria, four biological replicates and six technical replicates were grown in 24-well plates. Photographs were taken of each of the pellicles formed.

## Mucin binding ELISA

The ELISA binding assay was essentially performed as described by Rehman et al. (18) with both ChiC and ChiC$_{E143Q}$. Only mucin from porcine stomach type III was used (Sigma), and the substrate for detecting the horseradish peroxidase (HRP) activity of the anti-His-HRP-conjugated antibody, Ultra TMB ELISA Substrate solution, was used (Thermo Fisher Scientific). Fifty microliters of the Ultra TMB ELISA Substrate solution was added to the wells and incubated for 30 min, followed by the addition of 50 µL 2 M sulfuric acid to stop the reactions. The data were recorded at 450 nm using a Varioskan LUX multimode microplate reader (Thermo Fisher Scientific).

## Screening of *P. aeruginosa* ChiC activity in the absence or presence of mucin

Starter cultures of *P. aeruginosa* UCBPP-PA14 and *P. aeruginosa* PAO1 were grown in LB at 37°C with shaking overnight. For *P. aeruginosa* UCBPP-PA14, the starter cultures were diluted 50 times in 150 mL of LB and grown at 37°C with shaking (200 rpm). For *P. aeruginosa* PAO1, the starter cultures were diluted 50 times in 150 mL LB or in 150 mL LB to which 1% mucin from porcine stomach (Sigma: type III) was added. The mucin was sterilized as described by Yeung et al. in addition to being treated with UV light for 15 min (89). Cultures were grown at 37°C with shaking (200 rpm). After 2, 6, and 24 h, two samples of 2 mL from each culture were taken and centrifuged at 10,000 rpm for 5 min. All pellets were washed once with water and pelleted by centrifugation. The supernatants were filtered using a 0.22 µm filter. ChiC activity was measured by incubating the pellets and the supernatants with (GlcNAc)$_4$ (Megazyme, ≥95% purity). The pellets were incubated in 100 µL reaction mixtures with 20 mM Tris-HCl buffer pH 7.5 and 0.4 mM (GlcNAc)$_4$. For the supernatants, the mixtures contained 50 µL of filtered supernatant with the same concentrations of buffer and (GlcNAc)$_4$ in total volumes of 100 µL. All control reactions were performed without substrate addition. As a positive control, reactions containing 1 µM ChiC were incubated in 100 µL reaction mixtures with 20 mM Tris-HCl buffer pH 7.5 and 0.4 mM (GlcNAc)$_4$. The reactions were incubated at 37°C for 2 h with no shaking in a Thermomixer C (Eppendorf, Hamburg, Germany). After 2 h, the reactions containing the bacteria were centrifuged at max speed for 5 min. If necessary, the reactions were transferred to a new tube, throwing away the bacterial pellets, and stopped by adding H$_2$SO$_4$ to a final concentration of 5 mM. All reactions were run in biological triplicates. The samples were filtered using a MultiScreenHTS HV Filter Plate 0.45 µm (Millipore). (GlcNAc)$_2$ formation by ChiC was measured as described earlier, using a Dionex Ultimate 3000 HPLC system with a Rezex RFQ-Fast Acid H+ (8%) analytical column.

## SEC analysis of reactions incubated with ChiC and mucin extracts

Reaction mixtures (200 µL) containing 20 mM Tris-HCl buffer pH 7.0 and 1 µM ChiC or ChiC$_{E143Q}$ were incubated with suspensions of mucin from porcine stomach (Sigma: type II and type III) at a final concentration of 4 mg/mL. The samples were incubated overnight at 37°C with shaking (600 rpm) in a Thermomixer C (Eppendorf, Hamburg, Germany). In the control reactions, either the substrate or ChiC were not included. Following the

incubation, the samples were centrifuged at 16,900 × $g$ for 5 min in an Eppendorf 5418R centrifuge prior to boiling at 100°C for 5 min. All reactions were run in triplicate.

Sample analysis was performed using a Dionex Ultimate 3000 HPLC system with a 300 × 7.8 mm Yarra-SEC2000 analytical column operated at 25°C using 20 mM Tris-HCl buffer pH 7.0 as the eluent, with an isocratic flow of 0.75 mL/min. The run time was 45 min for each sample, and the injection volume was set to 40 µL. The samples were detected using UV light at wavelengths of 194, 205, and 280 nm.

## Murine model of IT infection

IT infection was performed as previously described with modifications (9). PAO1 and the *chiC*::ISphoA/hah transposon mutant were grown overnight in LB at 37°C and 200 rpm. The bacterial strains were washed two times with PBS, and the pellet was resuspended in PBS to yield a final concentration of 1.5 × $10^7$ CFU/30 µL/mice. Female CD-1 mice (Charles River Laboratories, 10 weeks old) were anesthetized with 100 mg kg$^{-1}$ ketamine and 10 mg kg$^{-1}$ xylazine and infected intratracheally with PAO1 ($n$ = 10 mice) or the *chiC*::ISphoA/hah transposon mutant ($n$ = 9 mice), and lethal events were recorded accordingly.

## Analysis of the bacterial proteome of PAO1 and PAO1 *chiC* transposon mutant

Starter cultures of WT and PAO1 *chiC* transposon mutant (strain PW4887; *chiC*::ISphoA/hah) were grown in LB at 37°C with shaking overnight. The starter cultures were diluted 50 times in RPMI supplemented with 10% LB. The bacteria were grown until they reached the early exponential phase. PMSF, PhosSTOP (Roche), and Complete Mini EDTA-free protease inhibitors (Roche) were then added to the samples with final concentrations of 1 mM, 1×, and 1×, respectively. The bacterial pellets and supernatants were separated by centrifugation (4,500 × $g$, 15 min, 4°C). The pellets were washed twice with ice-cold PBS and centrifuged, before being resuspended in lysis buffer containing 20 mM Tris-HCl (pH 7.5), 0.1 M NaCl, 1 mM EDTA, 1× Complete Mini EDTA-free protease inhibitors, and lysozyme (0.5 mg/mL). Cells were disrupted by sonication (20×, 5″ off–5″ on, 26% amp), and the cellular debris was cleared by centrifugation (4,500 × $g$, 30 min, 4°C). Protein extracts were stored at −80°C.

The protein samples were in-gel-digested, and peptides were desalted and cleaned as previously described (88).

The peptide samples (200 ng) were analyzed by coupling a nano-UPLC (nanoElute, Bruker) to a trapped ion mobility spectrometry and a quadrupole time of flight mass spectrometer (timsTOF Pro, Bruker). The peptides were separated by an Aurora Series 1.6 µm C18 reverse-phase 25 cm × 75 µm analytical column with nanoZero and CaptiveSpray Insert (IonOpticks, Australia). The temperature of the column was kept at 50°C using the integrated oven. Equilibration of the column was performed before sample loading (equilibration pressure 900 bar). The flow rate was set to 0.4 nL/min, and the samples were separated using a solvent gradient from 2% to 37% solvent B over 100 min. Afterward, the gradient was increased to 95% solvent B in 10 min and maintained at the same level for the final 10 min. In total, a run time of 120 min was used for the separation of the peptides. Solvent A consisted of 0.1% (vol/vol) formic acid in water and 99.9% water, while solvent B consisted of 0.1% (vol/vol) formic acid in water and 99.9% (vol/vol) acetonitrile.

The timsTOF Pro was run in the positive ion data-dependent acquisition PASEF mode with the control software Compass Hystar version 5.1.8.1 and timsControl version 1.1.19 68. The acquisition mass range was set to 100–1,700 m/z. The TIMS settings were 1/K0 start 0.6 V·s/cm$^2$ and 1/K0 end 1.6 V·s/cm$^2$, ramp time 100 ms, ramp rate 9.42 Hz, and duty cycle 100%. The capillary voltage was set at 1,500 V, dry gas at 3.0 L/min, and dry temp at 180°C. The MS/MS settings were the following: number of PASEF ramps 10, total cycle time 1.17 sec, charge range 0–5, scheduling target intensity 20,000, intensity threshold

2,500, active exclusion release after 0.4 min, and CID collision energy ranging from 20 to 59 eV.

The raw files were processed as described earlier, with minor changes. The peptides were searched against the *P. aeruginosa* (strain PAO1) (UniProt proteome: UP000002438). Valid values were filtered with a minimum of 3 values in at least one group, and for pairwise Student's *t*-test, the significantly regulated proteins were filtered using a *P*-value of 0.01. MaxQuant version 2.0.2.0 and Perseus version 1.6.15.0 were used.

For the KEGG enrichment analysis, the clusterProfiler package for R was used (90). Enriched KEGG pathways were calculated using a hypergeometric test, and the filtered proteins were used as a background for the enrichment. The *P*-values from the hyper-geometric calculation were subjected to Benjamini–Hochberg adjustment and FDR correction. Pathways having *q*-values ≤0.05 were defined as enriched.

## Protease activity assay

Protease activity was evaluated using casein, which is based on the method described previously by Fujii et al. (91) with some modifications. A sample of 0.5 mL of a 1% casein solution (adjusted to pH 7.5) was combined with buffer and enzyme to a concentration of 25 mM Tris pH 7.5 and 1 µM ChiC, or 2 mg/mL ProteinaseK (NEB), respectively. The samples were incubated at 37°C with shaking (800 rpm) overnight (approximately 20 h) in a Thermomixer C (Eppendorf, Hamburg, Germany). After the addition of 1 mL of the precipitation solution, containing 0.1 M trichloroacetic acid, 0.22 M sodium acetate, and 0.33 M acetic acid, the mixtures were incubated at room temperature for 2–3 min, followed by centrifugation at 3,000 rpm for 10 min. The negative controls were prepared by adding 1 mL of the precipitating solution to 0.5 mL of the casein solution, followed by the addition of 0.5 mL of the buffer and enzyme as the final step. The absorbance was measured using a Cary 60 UV-Vis Spectrophotometer (Agilent) at 275 nm. All reactions were performed in triplicate.

## ACKNOWLEDGMENTS

We would like to thank Prof. Nina van Sorge for providing the strain *Pseudomonas aeruginosa* UCBPP-PA14 and Assistant Prof. Lisa Racki for providing the *Pseudomonas aeruginosa* UCBPP-PA14 chiC::MAR2xT7 transposon mutants. We acknowledge the services provided by the Proteomics Core Facility at the Norwegian University of Life Sciences, Genscript Biotech Corp, the Manoil lab (91), and the Protein-Glycan Interaction Resource of the CFG and the National Center for Functional Glycomics (NCFG) at Beth Israel Deaconess Medical Center, Harvard Medical School. The structure prediction of ChiC was performed with the resources provided by UNINETT Sigma2—The National Infrastructure for High-Performance Computing and Data Storage in Norway, under project number NN1003K. We are grateful for the assistance of Dr. G. Mathiesen, Dr. M. Skaugen, Martin Larsen, and Stanley Davis.

This work was funded by the Norwegian University of Life Sciences (NMBU) and by the National Institute of Health (1U54HD090259).

P.K.T.E., F.A., and G.V.-K. designed the experiments and wrote the paper. P.K.T.E, F.A., and R.Z. performed the experiments. P.K.T.E., F.A., and G.V.-K. analyzed and interpreted the data. V.N. contributed to writing and experimental and intellectual input. All authors reviewed and approved the paper.

## AUTHOR AFFILIATIONS

[1]Faculty of Chemistry, Biotechnology and Food Science, Norwegian University of Life Sciences, Ås, Norway
[2]Division of Host-Microbe Systems & Therapeutics, Department of Pediatrics, UC San Diego School of Medicine, La Jolla, California, USA

³Skaggs School of Pharmacy and Pharmaceutical Sciences, UC San Diego, La Jolla, California, USA

## AUTHOR ORCIDs

Per Kristian Thorén Edvardsen ⓘ http://orcid.org/0009-0004-7747-122X
Victor Nizet ⓘ http://orcid.org/0000-0003-3847-0422
Gustav Vaaje-Kolstad ⓘ http://orcid.org/0000-0002-3077-8003

## FUNDING

| Funder | Grant(s) | Author(s) |
|---|---|---|
| Norges Miljø- og Biovitenskapelige Universitet (NMBU) | | Gustav Vaaje-Kolstad |
| HHS \| National Institutes of Health (NIH) | 1U54HD090259 | Victor Nizet |

## DATA AVAILABILITY

The proteomics data obtained in this study can be accessed in the PRIDE database (https://www.ebi.ac.uk/pride/) using the accession numbers PXD048320, PXD048596, and PXD048597.

## ADDITIONAL FILES

The following material is available online.

### Supplemental Material

**Supplemental figures (Spectrum00546-24-s0001.docx).** Fig. S1 to S11.
**Table S1 (Spectrum00546-24-s0002.xlsx).** Glycan array screening data.
**Table S2 (Spectrum00546-24-s0003.xlsx).** GlcNAc2 addition, all proteins detected.
**Table S3 (Spectrum00546-24-s0004.xlsx).** GlcNAc2 addition, significantly regulated proteins.
**Table S4 (Spectrum00546-24-s0005.xlsx).** Pel, all proteins detected.
**Table S5 (Spectrum00546-24-s0006.xlsx).** chiClSphoAhah-mutant vs WT, all proteins detected.
**Table S6 (Spectrum00546-24-s0007.xlsx).** chiClSphoAhah-mutant vs WT, significantly regulated proteins.
**Table S7 (Spectrum00546-24-s0008.xlsx).** chiClSphoAhah mutant vs WT, KEGG enrichment analysis of upregulated proteins.

### Open Peer Review

**PEER REVIEW HISTORY (review-history.pdf).** An accounting of the reviewer comments and feedback.

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
