## [Reviewer comments · Microbiology Spectrum]

Microbiology Spectrum

Exploring roles of the chitinase ChiC in modulating *Pseudomonas aeruginosa* virulence phenotypes

Per Kristian Edvardsen, Fatemeh Askarian, Raymond Zurich, Victor Nizet, and Gustav Vaaje-Kolstad

Corresponding Author(s): Gustav Vaaje-Kolstad, Norges miljø- og biovitenskapelige universitet

Review Timeline:

Submission Date:	February 29, 2024
Editorial Decision:	April 1, 2024
Revision Received:	April 22, 2024
Accepted:	April 23, 2024

Editor: Fikri Avci

Reviewer(s): The reviewers have opted to remain anonymous.

Transaction Report:

DOI: <https://doi.org/10.1128/spectrum.00546-24>

Re: Spectrum00546-24 (Exploring roles of the chitinase ChiC in modulating *Pseudomonas aeruginosa* virulence phenotypes)

Dear Dr. Vaaje-Kolstad:

Thank you for the privilege of reviewing your work. We have completed the review of your manuscript, and I am pleased to inform you that both reviewers were enthusiastic. However, acceptance will not be final until you have addressed the reviewer comments. Below are instructions from the Spectrum editorial office and the reviewer comments.

Revision Guidelines

Sincerely,
Fikri Avci
Editor
Microbiology Spectrum

Reviewer #1 (Comments for the Author):

This study concerns the potential roles of the chitin's ChiC in modulating *Pseudomonas aeruginosa* virulence phenotypes. It is overall a well planned study and the results are of interest to a wide readership. I have the following comments and suggestions for the manuscript:

Abstract, line 28: perhaps add "marginally" before "increased".

Results, line 79: what kind of processing?

Results, line 95: "M. marina" in italics

Results, line 130-133: Please, indicate common range for chitin's Km values

Figure 3: I like the way SD is shown

Figure 4C, line 187: In my opinion this is not obvious from the figure, also the trend is difficult to interpret from Fig. 4D (line 187-188)

Fig 4E and 4F (referred to in line 195-197): perhaps indicate by colouring the gene name/number which overall functional group it belongs to?

Line 239-246: This is difficult to read without understanding the relevant phenotype of the transposon insertion mutants. This information could be added to the Description column in the table of bacterial strains in the Materials and Methods section. Same comment regarding PA14 in line 286, although in that case it appear to be explained in line 288-289. And again, for the transposon insertion mutant mentioned in line 318.

Line 337-338: This sentence is not clear to me, please reword.

Discussion, line 383-394: In more general terms, can an enzyme that carry out a degradation of a substrate (regardless of whether it is a chitin's or have a putative protease activity) be perceived to play a role in biosynthesis of a biofilm? Examples on that would be of interest to mention. Also, regarding the putative protease activity of ChiC that is not something that can be examined to some degree by exploring the tertiary structure of the enzyme, including the supposed catalytic site?

Materials and methods, table with bacterial strains: Is Vmax an E. coli strain?

Reviewer #2 (Comments for the Author):

In their submitted manuscript, the authors characterize the *Pseudomonas aeruginosa* secreted chitinase ChiC. They show that ChiC exhibits activity against alpha and beta chitin, as well as chitooligosaccharides. However, *P. aeruginosa* can only use GlcNAc but not (GlcNAc)₂ as a carbon source for growth. The measured Km was rather high for a canonical chitinase, suggesting additional substrates for ChiC. The authors showed that ChiC binds to mucins and that the presence of mucins results in increased secretion of ChiC. Chitobiose produced by the activity of ChiC functions as a signaling molecule and upregulated genes associated with virulence. ChiC was also found as the most predominant protein in the Pel exopolysaccharide. Surprisingly, a chiC transposon mutant was more virulent in a mouse model than *P. aeruginosa* WT. The authors subsequently compared expression profiles between WT and mutant and showed that disruption of chiC resulted in upregulation of virulence genes, possibly explaining the in vivo phenotype.

The authors have performed highly rigorous experiments and presented and interpreted their data extremely carefully. Group sizes and statistical tests were appropriate. It was a pleasure reading this manuscript. Multiple open questions remain, but the authors carefully addressed these questions and appropriately stated where future experimentation will be required. The most interesting open question is whether mucins are cleaved by ChiC, as chitin is not a substrate that mammals produce. However, this does not constitute required experimentation, as the experiments themselves are sound. Only a couple concerns remain:

Major comments:

1) Growth of *P. aeruginosa* on M9+glucose is extremely limited and only grows to OD₆₀₀=0.175 in 24h. Growth in a 96 well plate without shaking as performed here does lead to lower growth than in a test tube, but this is extremely low. M9+glucose should serve as the positive control but with so little growth here, it is questionable if remaining panels are actually interpretable. If one of these substrates supported growth at 10% of that on glucose, it would not be detectable. The authors should either repeat these experiments in conditions that yield higher OD₆₀₀ or perform an endpoint growth curve, determining the final CFU/ml for each substrate after 24h.

2) The biofilm experiments are not quantified. Ideally, the authors would quantify these findings. The biofilm pictures in the supplement are difficult to interpret. Are there some wells that did not form a biofilm? Were there more such wells for the chiC mutant?

Minor:

- Line 75: comprising

- Line 378-380. Correct grammar to clarify the meaning of this sentence.

Response to reviewers

Revision of “Exploring roles of the chitinase ChiC in modulating *Pseudomonas aeruginosa* virulence phenotypes”

Reviewer #1

Reviewer comment: Abstract, line 28: perhaps add "marginally" before "increased".

Response: We agree with the reviewer and have added this to the abstract, but instead of marginally we used the word slightly.

Original text, line 27-29.

“An intratracheal challenge model of PA pneumonia, the *chiC*::ISphoA/hah transposon insertion mutant paradoxically showed increased virulence compared to the wild-type parent strain.”

Revised text.

“An intratracheal challenge model of PA pneumonia, the *chiC*::ISphoA/hah transposon insertion mutant paradoxically showed slightly increased virulence compared to the wild-type parent strain.”

Reviewer comment: Results, line 79: what kind of processing?

Response: Thank you for your comment, we see that this was phrased imprecisely in the original manuscript. In the study referred to (ref. 11 in the manuscript), the authors suggest that the 11 N-terminal amino acids could represent a signal peptide as they found ChiC in the culture supernatant lacking these amino acids, thus being proteolytically processed/cleaved. To make the wording more precise, we have amended the text by including the word “proteolytical”

Original text, line 78-79.

“...it is notable that the first 11 amino acids have been reported to undergo processing in earlier studies of ChiC (11, 22).”

Revised text.

“...it is notable that the first 11 amino acids have been reported to undergo proteolytical processing in earlier studies of ChiC (11, 22).”

Reviewer comment: Results, line 95: "*M. marina*" in italics.

Response: We thank the reviewer for pointing this out and updated the text accordingly.

Original text line 93-95.

“Additionally, analysis and comparison of the predicted structure of ChiC were conducted with that of *M. marina*'s chitinase....”

Revised text.

“Additionally, analysis and comparison of the predicted structure of ChiC were conducted with that of *M. marina*'s chitinase....”

Reviewer comment: Results, line 130-133: Please, indicate common range for chitin's Km values.

Response: We appreciate the Reviewer's suggestion. Comparison of the ChiC K_M value towards chitotetraose for other family GH18 endochitinases towards similar substrates would indeed be interesting. A deeper investigation of the literature revealed that the *Serratia marcescens* endochitinase SmChiC has a K_M of 4 and 80 μ M towards the chitotriose and chitotetraose analogs 4-MU-(GlcNAc)₂ and 4-MU-(GlcNAc)₃, respectively (<https://doi.org/10.1080/10242420500518482>). The endochitinase Ech30 from *Trichoderma atroviride* has a K_M value against 4-MU-(GlcNAc)₂ that was measured to be 149 μ M (<https://doi.org/10.1016/j.bbapap.2005.01.002>). Another example is the plant endochitinase hevamine which has a K_M value of 13.8 and 3.2 μ M against (GlcNAc)₅ and (GlcNAc)₆, respectively ([https://doi.org/10.1016/S0014-5793\(00\)01833-0](https://doi.org/10.1016/S0014-5793(00)01833-0)). Furthermore, a chitinase purified from a prawn (*Penaeus japonicus*) showed K_M values of 249, 18 and 5 μ M against (GlcNAc)₄, (GlcNAc)₅ and (GlcNAc)₆, respectively (<https://doi.org/10.1080/00021369.1990.10870347>). These values support our claim that the ChiC K_M value for chitotetraose is higher than what is usually observed for similar endochitinases. We have referred to the most relevant studies (those that report on chitotetraose or the chitotetraose analogue) in the revised version of the manuscript and rephrased the original text.

Original text, line 130-133.

"Notably, the K_M value is significantly higher than what is commonly reported for chitinases, suggesting that the enzyme may have evolved to bind and/or cleave other substrates"

Revised text.

"Notably, the K_M value is significantly higher than what is reported for other endo-chitinases, e.g. the prawn endo-chitinase shows a K_M of 249 μ M for chitotetraose (29) and the *S. marcescens* endo-chitinase SmChiC shows a K_M of 80 μ M for the chitotetraose analogue 4-MU-(GlcNAc)₃ (30). This suggests that ChiC may have evolved to bind and/or cleave other substrates."

Reviewer comment: Figure 3: I like the way SD is shown.

Response: Thank you.

Reviewer comment: Figure 4C, line 187: In my opinion this is not obvious from the figure, also the trend is difficult to interpret from Fig. 4D (line 187-188).

Response: When revisiting Figure 4, panels C and D, we agree that the related text is not optimally phrased. For further clarification, additional details has been added to the text.

Original text, line 186-188.

"Many of the top regulated proteins during the exponential growth phase were associated with virulence and adaption to the host environment (Fig. 4C) a trend further supported by STRING analysis (Fig. 4D)."

Revised text.

"For Figure 4C, we have only included proteins that have been previously associated with infection and virulence in the literature. By examining the string analysis network on the left-hand side (**Fig. 4D**), one can observe the clustering of the upregulated proteins depicted in Figure 4C, with the exception of ApeB, which was not found to be linked to any other proteins. "

Reviewer comment: Fig 4E and 4F (referred to in line 195-197): perhaps indicate by colouring the gene name/number which overall functional group it belongs to?

Response: We thank the reviewer for pointing this out and agree that this is a better way to do this. Accordingly, we decided to color the proteins names having the same GO term for 3 or more proteins.

Reviewer comment: Line 239-246: This is difficult to read without understanding the relevant phenotype of the transposon insertion mutants. This information could be added to the Description column in the table of bacterial strains in the Materials and Methods section. Same comment regarding PA14 in line 286, although in that case it appear to be explained in line 288-289. And again, for the transposon insertion mutant mentioned in line 318.

Response: We thank the reviewer for bringing this to our attention and acknowledge that a more comprehensive understanding of these transposon mutants is necessary for clarity on this matter. In the table of bacterial strains under the column called description we added the following sentence for the *chiC* transposon mutants: “Features a transposon in *chiC*, resulting in the inactivation the chitinase protein.” Furthermore, we added a sentence to the original text regarding the Pel biofilm for clarification.

Original text, line 239-246.

“Additionally, we tested the activity of the extracted Pel biofilms from the *chiC::MAR2xT7* transposon insertion mutants D7 and D11, previously described (50), on the 4MU-(GlcNAc)₂ substrate. Our results show that the D7 transposon mutant displayed no discernible activity against this substrate, whereas the D11 transposon mutant exhibited some degree of activity. The disparity in activity levels between the D7 and D11 transposon mutants may be attributed to the insertion of the transposon in distinct regions of *chiC*.”

Revised text.

“Additionally, we tested the activity of the extracted Pel biofilms from the *chiC::MAR2xT7* transposon insertion mutants D7 and D11, previously described (53), on the 4MU-(GlcNAc)₂ substrate. We anticipated that the extracted biofilms would exhibit no chitinase activity towards this substrate, as both insertion mutants harbor transposons within the gene encoding ChiC. Our results show that the D7 transposon mutant displayed no discernible activity against this substrate, whereas the D11 transposon mutant exhibited some degree of activity. The disparity in activity levels between the D7 and D11 transposon mutants may be attributed to the insertion of the transposon in distinct regions of *chiC*.”

Reviewer comment: Line 337-338: This sentence is not clear to me, please reword.

Response: We agree and have rephrased the sentence in the revised manuscript:

Original text, line 337-338.

“The upregulation of these proteins suggests potential compensation for the loss of ChiC by the transposon mutant strain. The downregulated proteins were not substantially enriched in any pathways.”

Revised text.

“The increased expression of proteins involved in pathogenicity and enriched pathways, such as biofilm formation, quorum sensing, and bacterial secretion systems, suggests that the bacterium is actively attempting to recover from the inactivation and loss of ChiC.”

Reviewer comment: Discussion, line 383-394: In more general terms, can an enzyme that carry out a degradation of a substrate (regardless of whether it is a chitin's or have a putative protease activity) be perceived to play a role in biosynthesis of a biofilm? Examples on that would be of interest to mention. Also, regarding the putative protease activity of ChiC that is not something that can be examined to some degree by exploring the tertiary structure of the enzyme, including the supposed catalytic site?

Response: We are grateful for the comment and the good questions. In short, such activity do exist. One example is PelA, which is a glycoside hydrolase important for the formation of the pellicle biofilm exopolysaccharide (Pel) in *P. aeruginosa*. The protein possesses two different activities against the carbohydrate part making up the biofilm. PelA both cleave and partially deacetylate the pellicle carbohydrate polymer (<https://doi.org/10.1128/jb.02150-12>). This has been shown to be essential for the formation of the pellicle biofilm (<https://doi.org/10.1128/jb.02150-12>). Based on the current structure of the manuscript, we find it difficult to incorporate this information into the discussion. As a result, we have respectfully decided not to do so.

The potential of ChiC to possess protease activity is an intriguing topic that could be explored by examining its tertiary structure and catalytic site. When comparing the ChiC structural model with the structure of the proteolytic chitinase ChiA from *Legionella pneumophila* (having protease activity) (See ref 18 in the manuscript), we observe that the only conserved residues between these two chitinases are the catalytic glutamates in the active sites (E143 in ChiC and E543 in ChiA). None of the other residues believed to be important for the protease activity of ChiA seem to be present in ChiC. However, it is crucial to acknowledge that the discovery of protease activity in chitinases is relatively recent and an unprecedented discovery (see ref 18 in the manuscript). The mechanisms underlying this phenomenon is yet to be fully explored. While chitinases have typically been thought to exhibit activity towards carbohydrates, recent discoveries have broadened our understanding of their roles in infection biology and demonstrated a variety of functions (see refs 18, 65, 66 in the manuscript). In order to assess ChiC's potential protease activity, we conducted an assay where we tested ChiC in a casein degradation assay (as described by Fujii et al. (1983) (<https://doi.org/10.1128%2Fjb.154.2.831-837.1983>)), using Proteinase K as control enzyme. After 20 hours of incubation at 37 °C, we could not detect any protease activity of ChiC against casein (see figure below, “neg. control” are samples immediately stopped after addition of enzyme). In contrast, Proteinase K exhibited evident protease activity. We have added this figure to the revised manuscript as supplementary figure S11.

Original text, line 394.

“However, further experiments are required to pursue this hypothesis.”

Revised text.

“However, no activity was observed against casein for ChiC (**Fig. S11**) but further experiments are required to pursue this hypothesis.”

Reviewer comment: Materials and methods, table with bacterial strains: Is Vmax an E. coli strain?

Response: Thanks for pointing this out. Vmax cells are chemically competent cells of *Vibrio natriegens* which have emerged as a possible next-generation platform for bacterial expression of recombinant proteins. We further clarified this in the table listing the used bacterial strains in the study.

Original text.

“Vmax X2 pET28b SmChiC”

Revised text.

“*Vibrio natriegens* Vmax X2 pET28b SmChiC”

Reviewer #2

Reviewer comment: Line 75: comprising.

Response: Thank you for your feedback and comment. We have corrected the grammatical error that were previously present.

Reviewer comment: Line 378-380. Correct grammar to clarify the meaning of this sentence.

Response: Thank you for your comment. We have corrected and rewritten the sentence for clarification.

Original text, line 378-382.

“Thus, it is not unlikely that the bacterium will be exposed to this dimeric sugar in a host context. It is also possible that mucin glycans composing of carbohydrate motifs that are recognized by ChiC given the binding and possible activity towards porcine gastric mucins (**Fig. 6A; Figs. S9 and S10**). As the porcine mucin glycans only represent a model substrate, experiments using human mucins would be of great value to test the validity of the present results.”

Revised text.

“Thus, it is not unlikely that the bacterium may encounter this dimeric sugar in a host environment, taking into account its binding and potential activity towards porcine gastric mucins (**Figure 6A; Figures S9 and S10**). While porcine gastric mucin serves as a beneficial model substrate, using human mucins would offer better insights into the validity of the current findings.”

Reviewer comment: Growth of *P. aeruginosa* on M9+glucose is extremely limited and only grows to OD600=0.175 in 24h. Growth in a 96 well plate without shaking as performed here does lead to lower growth than in a test tube, but this is extremely low. M9+glucose should serve as the positive control but with so little growth here, it is questionable if remaining panels are actually interpretable. If one of these substrates supported growth at 10% of that on glucose, it would not be detectable. The authors should either repeat these experiments in conditions that yield higher OD600 or perform an endpoint growth curve, determining the final CFU/ml for each substrate after 24h.

Response: Very good point made by the reviewer. In response, we have now included new growth curves performed with shaking to obtain better growth. As expected, we observe a more than doubling of OD600 in M9 medium containing glucose (OD600 ~0.3). These experiments were conducted using four biological replicates, and shaking was performed every 5 minutes. Importantly, the growth curves obtained for GlcNAc, GalNAc and (GlcNAc)₂ were essentially identical to what was observed in the original experiment. Corroborating these data, a study by Johnson et al. (2008) (<https://doi.org/10.1371/journal.pgen.1000211>) demonstrated, using Biolog microarrays, that *P. aeruginosa* PAO1 can utilize GlcNAc as a carbon source, but not GalNAc (as stated in the original manuscript). Furthermore, we grew bacteria overnight in 4 mL of M9 with either glucose, GlcNAc, (GlcNAc)₂ or GalNAc. The bacteria after approximately 20 h of growth had an OD600 of 0.428, 0.085, 0.018 and 0.037, respectively, indicating growth in glucose, slight growth in GlcNAc and no growth in (GlcNAc)₂ and GalNAc as our microtiter growth curve assays show in Figure 3A-E. Based on these findings, we are confident that our results are accurate.

Reviewer comment: The biofilm experiments are not quantified. Ideally, the authors would quantify these findings. The biofilm pictures in the supplement are difficult to interpret. Are there some wells that did not form a biofilm? Were there more such wells for the *chiC* mutant?

Response: We concur with the reviewer that it would be ideal to quantify the biofilm in order to determine any differences between the WT and the *chiC* transposon mutant. However, the pellicle biofilm that we are working with is not easily quantifiable using standard biofilm assays. The crystal violet assay, which is commonly used for biofilm quantification, is not suitable for pellicle biofilms because the biofilm is formed on the air-liquid surface and is lost when the media is discarded, and the wells are washed. Additionally, we attempted to extract the biofilm and resuspend it in tetrahydrofuran, as described by Syal (2017) (<https://doi.org/10.1007/s00284-017-1304-0>), but this method was not applicable for *P. aeruginosa*. There is currently a lack of sufficient methodology for conducting experiments with pellicle biofilms. Therefore, we took pictures of the pellicles formed on the air-liquid surface for the bacteria. This is a common practice for the bacterium *Bacillus subtilis*, which also forms pellicle biofilms. An example of this is seen in the paper by Kobayashi (2007) (<https://doi.org/10.1128/jb.00157-07>). For both the WT and the *chiC* transposon mutant, we were able to observe the formation of pellicle biofilms in all the wells. No profound differences were observed between the two strains.

Re: Spectrum00546-24R1 (Exploring roles of the chitinase ChiC in modulating *Pseudomonas aeruginosa* virulence phenotypes)

Dear Dr. Vaaje-Kolstad:

Congratulations! Your manuscript has been accepted, and I am forwarding it to the ASM production staff for publication. Your paper will be checked to ensure all elements meet the technical requirements. ASM staff will contact you if anything needs to be revised before copyediting and production can begin. Otherwise, you will be notified when your proofs are ready to be viewed.

Sincerely,
Fikri Avci
Editor
Microbiology Spectrum